# FROM PIXELS TO TOKENS: BYTE-PAIR ENCODING ON QUANTIZED VISUAL MODALITIES

**Wanpeng Zhang**[1][*], **Zilong Xie**[2], **Yicheng Feng**[1], **Yijiang Li**[3], **Xingrun Xing**[4,5],
**Sipeng Zheng**[4], **Zongqing Lu**[1,6][†]

[1]School of Computer Science, Peking University
[2]The Chinese University of Hong Kong
[3]University of California, San Diego
[4]Beijing Academy of Artificial Intelligence
[5]Institute of Automation, Chinese Academy of Sciences
[6]BeingBeyond

## ABSTRACT

Multimodal Large Language Models have made significant strides in integrating visual and textual information, yet they often struggle with effectively aligning these modalities. We introduce a novel image tokenizer that bridges this gap by applying the principle of Byte-Pair Encoding (BPE) to visual data. Unlike conventional approaches that rely on separate visual encoders, our method directly incorporates structural prior information into image tokens, mirroring the successful tokenization strategies used in text-only Large Language Models. This innovative approach enables Transformer models to more effectively learn and reason across modalities. Through theoretical analysis and extensive experiments, we demonstrate that our BPE Image Tokenizer significantly enhances MLLMs' multimodal understanding capabilities, even with limited training data. Leveraging this method, we develop `Being-VL-0`, a model that demonstrates superior performance across various benchmarks and shows promising scalability, potentially paving the way for more efficient and capable multimodal foundation models. For further details, visit our website.

## 1 INTRODUCTION

The development of Multimodal Large Language Models (MLLMs) has made significant progress (Yin et al., 2023; Team et al., 2023; Liu et al., 2024b). However, these multimodal foundation models often model different modalities separately, incorporating many modality-specific designs such as specialized encoders and decoders (Liu et al., 2024b; Zhang et al., 2024; Jin et al., 2023). While this approach allows training data to align well with these modality-specific designs, it often struggles to achieve a unified understanding of multimodal information (Team, 2024). The primary reason for this limitation could be that while encoders of other modalities can learn rich information, without the assistance of the corresponding decoders, LLMs cannot fully comprehend the complex patterns contained within the embeddings provided by the encoder. In other words, LLMs need to learn to interpret the token embeddings again, which is the job of the decoders of other modalities, leading to difficulties in aligning with these modalities (Baltrušaitis et al., 2018).

Recent research has begun to explore unified token-based representations for MLLMs (Team, 2024; Lu et al., 2024; Zheng et al., 2024), attempting to achieve better capabilities for multimodal information processing by moving away from modality-specific designs. This approach offers two main advantages: first, it can achieve unified information understanding, and second, it enables unified generation that can be decoded into different modalities. However, these methods simply convert various modal information into unified representations and then directly feed them into Transformer models (Vaswani, 2017), hoping to learn the data just with massive computing resources. Although

---

[*]Wangpeng Zhang, Yicheng Feng, and Yijiang Li work as interns at BAAI
[†]Correspondence to Zongqing Lu <zongqing.lu@pku.edu.cn>.

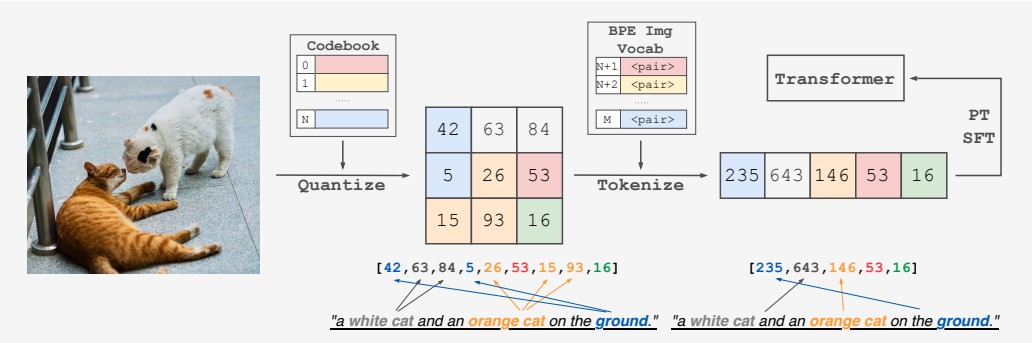

Figure 1: Illustration of our BPE Image Tokenizer. The overall process begins with quantizing the image into initial token IDs. The BPE Image Tokenizer then combines these tokens based on learned patterns, similar to text tokenizers. This combination results in tokens that inherently contain more semantic information. The final tokenized sequence thus incorporates structural prior information from the image, enabling the Transformer model to deeper comprehend the alignment between visual and textual information during training. This approach facilitates more effective integration of visual data into MLLMs, enhancing their multimodal understanding capabilities.

this approach can be effective to some extent, it consumes enormous computing resources and time while failing to achieve significant improvements in performance.

In comparison to the widely adopted training paradigm of text-only LLMs (Touvron et al., 2023; Achiam et al., 2023), we observe that current token-based MLLMs often overlook a crucial component: the tokenizer that explicitly merges data sources. In text-only LLMs, the most widely adopted tokenizer is the Byte-Pair Encoding (BPE) tokenizer (Sennrich, 2015; Radford et al., 2019), which learns to merge characters into tokens of varying lengths based on the word frequency in the training corpus before providing them to the Transformer for learning. While many believe this approach is solely for conserving computational resources, some recent theoretical and experimental analyses suggest that using merged tokens plays a vital role in the Transformer's learning of sequential data (Rajaraman et al., 2024; Makkuva et al., 2024; Merrill et al., 2024). In fact, if raw sequence data is provided directly, Transformer models may even fail to learn how to predict entirely.

In this paper, we reveal that Transformer models cannot effectively learn certain types of two-dimensional sequence data. However, when we apply operations similar to BPE tokenizers to the two-dimensional data, *i.e.*, merging tokens based on the frequency of training data, the learning effectiveness can improve significantly. We believe the benefit comes from the addition of necessary structural prior information to the tokens during the merging process, making it easier for Transformer models to be aware of important relational information in the data during learning. Our theoretical analysis proves that tokenizing sequence data on certain types of two-dimensional data can indeed achieve smaller losses, which we further validate experimentally.

Based on the insights provided by our theoretical analysis and experimental validation, we propose a new learning paradigm for MLLMs. By tokenizing the unified representation of multimodal data using a novel BPE Image Tokenizer, we enable Transformer models to better understand image data, allowing MLLMs built from text-only LLMs to have good multimodal capabilities. As illustrated in Figure 1, after quantizing the image into token IDs, the BPE Image Tokenizer further combines these token IDs according to its learned patterns. Similar to text tokenizers, our BPE Image Tokenizer ensures that the tokens being fed to the language model decoder inherently contain more semantically meaningful information in the statistical sense. Intuitively, the tokenized IDs, which have incorporated structural prior information from the image, could provide the Transformer model with better comprehension of the specific alignment between text and image in the training data. Preliminary results further validate the effectiveness of explicit tokenization for multimodal data. We hope that the new paradigm for learning MLLMs proposed in this paper will provide better guidance for subsequent scaling-up efforts, leading to the building of more powerful MLLMs.

In summary, we make the following key contributions:

- We are the first to propose a new MLLM learning paradigm that explicitly tokenizes multimodal data like text-only LLMs, centered around our novel BPE Image Tokenizer.
- We theoretically analyze why this learning paradigm can bring benefits and further provide corresponding experimental validation.
- We design an algorithm for training the BPE Image Tokenizer and train the `Being-VL-0` with this tokenizer. The performance evaluation further validates the capability enhancements this learning paradigm brings to MLLMs.

## 2 NOTATIONS AND FORMULATION

Before delving into the theoretical analysis of our proposed paradigm, we introduce key notations and concepts used throughout this paper.

**Image Representation and Quantization.** We represent an image as a set of patches $\mathcal{X} = \{X_{ij}\}_{1 \leq i,j \leq m}$, where $m$ is the number of patches per row/column, and $X_{ij}$ denotes the patch at position $(i, j)$. We employ Vector Quantization (VQ) to discretize these patches, using a codebook $\mathcal{C}$ with size $C = |\mathcal{C}|$, and a quantization function $\text{VQ} : \mathbb{R}^d \to \mathcal{C}$.

**BPE Image Tokenizer.** Our proposed BPE Image Tokenizer converts a quantized image into a sequence of token IDs, defined as $\text{enc} : \mathcal{X} \to \mathcal{T}^*$, where $\mathcal{T}$ is the set of tokens and $\mathcal{T}^*$ is the set of all finite sequences of tokens. A special case of encoding is flattening, $\text{flat} : \mathcal{X} \to \mathcal{C}^{m^2}$, which arranges image patches into a sequence row by row.

**Unigram Model.** For a unigram model $Q \in \mathcal{Q}_{\text{1-gram}}$, given a token sequence $\mathbf{t} = (t_1, ..., t_{|\mathbf{t}|})$, the probability is defined as:

$$Q(\mathbf{t}) = Q_{\#}(|\mathbf{t}|) \prod_{r=1}^{|\mathbf{t}|} Q_{\text{tok}}(t_r), \tag{1}$$

where $Q_{\#}$ is a distribution over $\mathbb{N}$ representing the probability of the sequence length, and $Q_{\text{tok}}$ is a distribution over the dictionary of tokens $\mathcal{T}$.

**Multimodal Large Language Model (MLLM).** We define an MLLM as a probabilistic model $Q$ over token sequences, capable of processing both text and image data as follows:

1. Input Processing: $\mathbf{t}_{\text{text}} = \text{tokenize}(\mathbf{x}_{\text{text}})$; $\mathbf{t}_{\text{image}} = \text{enc}(\text{VQ}(\mathbf{x}_{\text{image}}))$.
2. Sequence Modeling: $P(t_i|t_1, ..., t_{i-1}) = Q(t_i|\mathbf{t}_{<i})$.
3. Generation: $y_i \sim P(t_i|y_1, ..., y_{i-1}, \mathbf{t})$

The training objective for an MLLM is to minimize the cross-entropy loss:

$$\mathcal{L} = -\mathbb{E}_{(\mathbf{x},\mathbf{y}) \sim D} \left[ \sum_{i=1}^{|\mathbf{y}|} \log P(y_i|y_{<i}, \mathbf{t}) \right], \tag{2}$$

where $D$ is a dataset of input-output pairs $(\mathbf{x}, \mathbf{y})$. The goal is to find the optimal parameters $\theta^*$ that minimize this loss: $\theta^* = \arg\min_\theta \mathcal{L}(\theta)$.

## 3 THEORETICAL ANALYSIS

Previous studies have demonstrated that Transformers struggle to effectively learn certain one-dimensional sequence data (Rajaraman et al., 2024; Makkuva et al., 2024). In this section, we extend this concept to two-dimensional image data, considering a simplified model where the image data-generating distribution follows a 2D $k^{th}$-order Markov process. This simplified model is defined as follows:

**Definition 1** (2D $k^{th}$-order Markov process). *For each variable $X_{i,j}$, with probability $\frac{1}{2}$, it depends only on $X_{i-k,j}$, i.e., $Pr(X_{i,j} = 1|X_{i-k,j} = 0) = p$ and $Pr(X_{i,j} = 1|X_{i-k,j} = 1) = 1 - q$; With probability $\frac{1}{2}$, it depends only on $X_{i,j-k}$, i.e., $Pr(X_{i,j} = 1|X_{i,j-k} = 0) = p$ and $Pr(X_{i,j} = 1|X_{i,j-k} = 1) = 1 - q$. Figure 2(a) illustrates this process.*

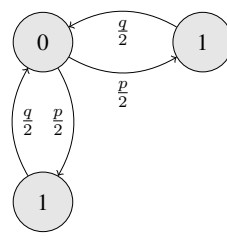
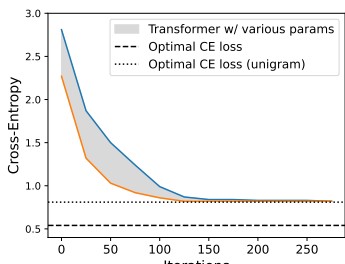
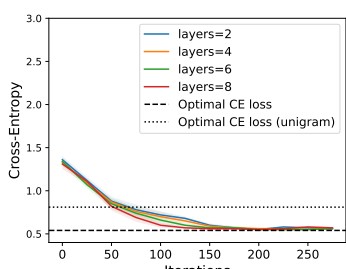

(a) 2D $k^{th}$-order Markov process. Sequence data is transformed based on horizontal or vertical $k^{th}$-order Markov conditional probabilities.

(b) Without tokenizer, Transformer fails to learn 2D $k^{th}$-order Markov sequence data, only achieving the performance of a unigram model (dotted line). The gray area represents Transformer under various parameters

(c) With BPE tokenizer, even Transformer models with only a few layers (less parameters) can easily learn the 2D $k^{th}$-order Markov sequence data, achieving optimal cross-entropy loss (dashed line).

Figure 2: Definition of 2D $k^{th}$-order Markov sequence data, and the performance of Transformer in learning such sequence data with or without tokenizer. For details of the hyperparameters used in the experiments, please refer to section B.

This simplification is intuitive, as pixels in an image often exhibit conditional dependencies with other pixels at specific horizontal and vertical distances. Consequently, real-world image data can be viewed as a composite of multiple such simplified processes. When attempting to learn such sequence data directly using a Transformer model, an interesting phenomenon occurs: the Transformer fails to surpass the performance of the stationary unigram model, regardless of various hyperparameter choices. As shown in Figure 2(b), the Transformer fails to improve the cross-entropy loss beyond that of the optimal unigram model on the training data (indicated by the dotted line). However, as shown in Figure 2(c), when encoding the sequences in each direction separately using BPE Tokenizer trained on this distribution, even a minimal Transformer model with only a few layers can readily achieve optimal loss. This finding suggests that BPE Tokenizer could play a crucial role in the Transformer's learning process for the defined two-dimensional sequence data.

To formally analyze this phenomenon in a more general setting, we first let $m$ be the number of patches per row in an square image, and let $X_{ij}$ denote the patch at position $(i, j)$ for $1 \leq i, j \leq m$. An image can be represented by the set of $m^2$ patches $\{X_{ij}\}_{1 \leq i,j \leq m}$, which we abbreviate as $\boldsymbol{X}$ for simplicity. Each patch can be encoded into an index with a VQ-GAN model's codebook, denoted by $\mathcal{C}$, with $C = |\mathcal{C}|$ representing the size of the codebook.

We then consider two scenarios for image generation:

**Scenario 1.** *Every column in the image $\boldsymbol{X}$ is generated independently by an ergodic Markov process supported on $\mathcal{C}$, proceeding from top to bottom, with the same transition kernel $P(\cdot \mid \cdot)$ and stationary distribution $\pi$.*

**Scenario 2.** *Every row in the image $\boldsymbol{X}$ is generated independently by an ergodic Markov process supported on $\mathcal{C}$, proceeding from left to right, with the same transition kernel $P(\cdot \mid \cdot)$ and stationary distribution $\pi$.*

Given a model $Q$ on tokens and an encoding function $\mathrm{enc}(\cdot)$ that encodes an image into a sequence of tokens, we define the cross entropy loss as

$$\mathcal{L}_m(Q \circ \mathrm{enc}) = -\frac{1}{m^2} \mathbb{E}[\log(Q(\mathrm{enc}(\boldsymbol{X})))],$$

where the subscript $m$ indicates the dependence on size $m$. We also introduce the trivial encoding function $\mathrm{flat}(\cdot)$, which merely flattens an image into a sequence of patches arranged row by row from left to right:

$$\mathrm{flat}(\boldsymbol{X}) = (X_{11}, X_{12}, ...X_{1m}, X_{21}, X_{22}, ..., X_{2m}, ..., X_{m1}, X_{m2}, ..., X_{mm}).$$

Using this notation, $\mathcal{L}_m(Q \circ \mathrm{flat})$ represents the cross entropy loss for model $Q$ along with the encoding function $\mathrm{flat}(\cdot)$. For simplicity, we abbreviate it as $\mathcal{L}_m(Q)$ throughout this paper. We

define $H(p)$ as the Shannon entropy for any discrete distribution $p$:

$$H(p) = -\sum_{y \in \mathrm{supp}(p)} p(y) \log p(y).$$

Now, we present our main theoretical results:

**Proposition 1.** *For data generating processes described in either Scenario 1 or Scenario 2, as $m \to \infty$, the optimal cross-entropy loss among unigram model family $\mathcal{Q}_{\text{1-gram}}$ satisfies*

$$\liminf_{m \to \infty} \min_{Q \in \mathcal{Q}_{\text{1-gram}}} \mathcal{L}_m(Q) \geq H(\pi) = \sum_{a \in \mathcal{C}} \pi(a) \log(\pi(a)). \tag{3}$$

*In contrast, the optimal unconstrained cross entropy loss satisfies*

$$\lim_{m \to \infty} \min_Q \mathcal{L}_m(Q) = H_\infty \overset{\triangle}{=} -\sum_{a \in \mathcal{C}} \sum_{a' \in \mathcal{C}} \pi(a) P(a' \mid a) \log\left(P(a' \mid a)\right). \tag{4}$$

Here we provide a sketch of the proof of Proposition 1. For the full proof, please see Appendix A.2.

*Sketch of Proof.* For the optimal cross-entropy loss among $\mathcal{Q}_{\text{1-gram}}$, observe that under both scenarios, the marginal distribution of each patch is precisely the stationary distribution $\pi$. When applying a unigram model to $\mathrm{flat}(\boldsymbol{X})$ and leveraging its inherent independence, the optimal loss closely approximates $H(\pi)$, with a discrepancy that vanishes as $m \to \infty$. Regarding the optimal unconstrained cross-entropy loss, note that the best model is simply the generating model of $\mathrm{flat}(\boldsymbol{X})$, with its cross-entropy loss converging to $H_\infty$ as $m \to \infty$. $\qquad\square$

**Remark 1.** *Consider the 2D $k^{th}$-order Markov process in Figure 2(a). Let $\frac{p}{2} = \frac{q}{2} = \frac{1-\delta}{2}$, which means switching between $0$ and $1$ with equal probability $p = q = 1 - \delta$. For this process, $\lim_{m \to \infty} \frac{1}{m} H(P) = \delta \log(\frac{1}{\delta}) + (1-\delta) \log(\frac{1}{1-\delta})$. However, the unigram stationary distribution is $\pi = \{\frac{1}{2}, \frac{1}{2}\}$, and so $H(\pi) = \log(2)$. The ratio $\lim_{m \to \infty} \frac{H(\pi)}{\frac{1}{m} H(P)}$ approaches $\infty$ as $\delta \to 0$. This example shows that the gap between $H(\pi)$ and $H_\infty$ could be arbitrarily large.*

The implications of Proposition 1 and Remark 1 are profound for understanding the limitations of simple models in capturing the structure of two-dimensional image data. This result shows a significant performance gap between unigram models and the optimal unconstrained model for such data. This gap, represented by the difference between $H(\pi)$ and $H_\infty$, can be arbitrarily large. This finding underscores a critical limitation of simple unigram models, which Transformers could default to when processing raw sequences, according to our earlier experimental results. These models are inherently restricted in their ability to capture the true structure of image data, suggesting that more sophisticated tokenization methods are necessary for effectively processing two-dimensional image data with Transformer models. We further discuss the potential information loss of our tokenizer in Appendix A.1.

**Proposition 2.** *For data generating processes described in either Scenario 1 or Scenario 2, assume that $\delta \overset{\triangle}{=} \min_{a,a' \in \mathcal{C}} P(a'|a) > 0$. Then there exists a tokenizer with a dictionary containing at most $D$ tokens, along with an encoding function $\mathrm{enc}(\cdot)$ applied to $\boldsymbol{X}$, such that*

$$\limsup_{m \to \infty} \min_{Q \in \mathcal{Q}_{\text{1-gram}}} \mathcal{L}_m(Q \circ \mathrm{enc}(\cdot)) \leq \frac{1}{1 - \varepsilon} H_\infty, \tag{5}$$

*where $\varepsilon = \log(1/\delta)/(0.99 \log(D))$ and $D \in \mathbb{N}$ is an arbitrary constant that is sufficiently large.*

Here we provide a sketch of the proof of Proposition 2. For the full proof, please see Appendix A.2.

*Sketch of Proof.* The proof leverages the fact that in both scenarios, the generated image is composed of $m$ parallel Markov sequences, each of length $m$, sharing the same transition kernel and stationary distribution. We first apply the result from the one-dimensional case (Lemma A.1) to each column (or row) separately. Then, we construct a tokenizer for the entire image by concatenating the tokens from each column (or row). By carefully bounding the cross-entropy loss of this constructed tokenizer, we show that it satisfies the desired inequality. The key step is to handle the additional complexity introduced by the two-dimensional structure while maintaining the bound derived from the one-dimensional case. $\qquad\square$

---

**Algorithm 4.1** BPE Image Tokenizer training procedure.

---

1:  Input $v_0, m, D$.                                  ▷ $v_0$: initial vocab size, $m$: new vocab size, $D$: training data
2:  $v \leftarrow v_0$                                          ▷ $v$: current vocab size
3:  $A \leftarrow \text{zeros}(v \times v)$                    ▷ $A$: adjacency matrix
4:  $V \leftarrow \emptyset$                                    ▷ $V$: extended vocabulary
5:  **for** $i \leftarrow 1$ to $m$ **do**
6:      $A \leftarrow \text{UpdateMatrix}(D)$
7:      $(p, f) \leftarrow \text{MaxFreqPair}(A)$          ▷ $p$: best pair, $f$: frequency
8:      **if** $f = 0$ **then break**
9:      **end if**
10:     $V \leftarrow V \cup \{(p, v)\}$
11:     $D' \leftarrow \emptyset$
12:     **for each** $d \in D$ **do**
13:         $d' \leftarrow$ Replace $p$ with $v$ in $d$
14:         $D' \leftarrow D' \cup \{d'\}$
15:     **end for**
16:     $D \leftarrow D'$
17:     $v \leftarrow v + 1$                                    ▷ set next id for new token
18: **end for**
19: **return** $V$

---

Proposition 2 offers insights into the potential benefits of using a tokenizer for image data. This result demonstrates that with an appropriate tokenizer, we can achieve a loss arbitrarily close to the optimal unconstrained loss $H_\infty$, significantly outperforming the unigram model bound of $H(\pi)$. These theoretical results motivate the design of our BPE Image Tokenizer. In the following sections, we will describe the implementation of our algorithm and present corresponding experimental results.

# 4 PRELIMINARY IMPLEMENTATION

To validate the effectiveness of our proposed BPE Image Tokenizer and the training paradigm in enhancing models' multimodal understanding capabilities, we implemented the entire procedure and conducted a series of experiments. Given the scope of this paper, we emphasize that this implementation for training an MLLM is preliminary and does not involve extensive scaling in terms of dataset size or computational resources to directly compare with state-of-the-art foundation models.

## 4.1 BPE IMAGE TOKENIZER TRAINING

**Dataset Preparation:** We constructed a diverse image dataset comprising 2.78 million images from ImageNet (Deng et al., 2009), CC (Sharma et al., 2018), LAION (Schuhmann et al., 2022), and SBU (Ordonez et al., 2011). We applied a filtering mechanism similar to LLaVA (Liu et al., 2024b), selecting images based on their class labels or noun-phrase statistics derived from captions to keep the concept diversity.

**Image Preprocessing:** We used a pretrained VQ-GAN model released by Chameleon (Team, 2024) to preprocess the images. This VQ-GAN model uses a codebook size of 8192 and quantizes images of any size into a 1024-dimensional ID tensor.

**Tokenizer Training Algorithm:** We developed the BPE image tokenizer training algorithm, inspired by the text BPE algorithm. The main procedure of this algorithm is presented in Algorithm 4.1, with supporting functions detailed in Algorithms B.1 and B.2 (see Appendix B). Key designs of our algorithm include: 1) Use of an adjacency matrix to count token co-occurrences; 2) Simultaneous consideration of horizontal and vertical token combinations; 3) Ability to merge tokens into arbitrary shapes through iterative merging.

**Extended Vocabulary:** The training process results in an extended vocabulary that merges tokens based on learned patterns. To analyze the impact of vocabulary size on performance, we trained multiple tokenizer versions with extended vocabulary sizes ranging from 1024 to 16384.

## 4.2 MLLM TRAINING

**Base Model:** We used the `Llama-3.1-8B` model (Dubey et al., 2024) as our base text LLM.

**Token Embedding Expansion:** Before training the MLLM, we first expanded the token embedding layers of the base model to accommodate the new image token IDs. For instance, with 8192 VQ-GAN token IDs and a BPE Image Tokenizer vocabulary of size 4096, we expanded the embedding layers from $n \times m$ to $(n + 8192 + 4096) \times m$, where $n$ is the original text embedding dimension and $m$ is the number of embedding layers. We also added extra special tokens to mark the beginning and end positions of images.

**Training Process:** The training process consisted of two stages:

- *Stage 1: Image Understanding Pretraining (PT).* In this stage, we froze the original text token embeddings and trained only the new image token embeddings using image caption data. This stage used 595K images from CC-3M (Sharma et al., 2018) and 558K from LCS (Liu et al., 2024b).
- *Stage 2: Supervised Fine-Tuning (SFT).* After stage 1, we unfroze all weights and performed full-parameter fine-tuning using complex conversation data with image inputs. This stage used 1.27 million entries from the LLaVA-OneVision Dataset (Li et al., 2024), including General QA (504K), Doc & Chart (249K), Reasoning (343K), and OCR (180K) tasks.

The core distinction between our proposed training pradigm and existing widely used MLLM training approaches (Liu et al., 2024b;a) mainly lies in the first stage of pretraining. The conventional approaches typically align a pretrained CLIP encoder (Radford et al., 2021) with a text-based LLM, where the image understanding capability is largely derived from the CLIP encoder, which was trained on 400 million image-text pairs (Radford et al., 2021), far exceeding the amount of data we provide to our model for image understanding. Though our proposed paradigm also involves pretraining an image tokenizer, it does not directly provide image comprehension capability to the tokenizer. Instead, the MLLM acquires its entire image comprehension capability after the first stage of training. Intuitively, this approach allows for a more direct fusion of the image modality with the text-based LLM, potentially mitigating biases introduced by separate training and subsequent alignment. Our experimental results also can demonstrate that even with a significantly limited amount of data compared to that used in pretraining the CLIP encoder, we can still endow text-based LLMs with good image understanding capabilities.

## 5 EXPERIMENTAL RESULTS AND ANALYSIS

Our experiments aimed to evaluate the effectiveness of the proposed BPE Image Tokenizer and its impact on MLLM performance. We analyze the results from three main perspectives: the impact of the BPE Image Tokenizer, the effect of additional data scaling, and the influence of extended vocabulary. Table 5.1 summarizes our main results, while Figures 3(a) and 3(b) provide insights into the impact of the BPE vocabulary and its token usage patterns.

### 5.1 EXPERIMENTS SETUP

**Benchmarks:** We evaluated our model using multiple benchmarks:

- VQAv2 (Goyal et al., 2017): Visual Question Answering
- MMBench (Liu et al., 2023): Multimodal Understanding
- MME (Fu et al., 2023): Multimodal Evaluation (separate tests for perception $MME^p$ and cognition $MME^c$)
- POPE (Li et al., 2023): Object Hallucination Evaluation
- VizWiz (Gurari et al., 2018): Visual Question Answering for Visually Impaired Users

**Data Scaling Experiments:** To further investigate the impact of dataset size on performance, we gradually augment our training data with RefCOCO (50.6K) (Kazemzadeh et al., 2014), AOKVQA (66.2K) (Schwenk et al., 2022), ShareGPT4o (57.3K) (Chen et al., 2023), and ALLaVA Inst (70K) (Chen et al., 2024).

Table 5.1: Performance comparison of different settings and training strategies across multiple benchmarks. Consistent improvements across all benchmarks demonstrate the benefit from the BPE Image Tokenizer and the effectiveness of the designed training strategy. Further performance gains are observed with incremental data additions, highlighting the potential scalability of our approach. Specifically, the 'LLM' represents the `Llama-3.1-8B` backbone model we used. $MME^p$ and $MME^c$ represent MME-perception and MME-cognition, respectively.

| | Training type | VQAv2 | MMBench | $MME^p$ | $MME^c$ | POPE | VizWiz |
|---|---|---|---|---|---|---|---|
| LLM+VQ | SFT | 51.1 | 35.9 | 972.3 | 231.8 | 73.8 | 43.1 |
| | PT(full)+SFT | 53.7 | 37.0 | 1037.2 | 261.4 | 75.3 | 44.2 |
| | PT(freeze)+SFT | 55.4 | 37.6 | 1054.5 | 277.0 | 76.0 | 45.3 |
| LLM+VQ+BPE (`Being-VL-0`) | SFT | 52.2 | 35.4 | 1029.7 | 269.6 | 76.3 | 45.3 |
| | PT(full)+SFT | 56.5 | 38.6 | 1144.6 | 284.3 | 77.3 | 45.8 |
| | PT(freeze)+SFT | 57.1 | 40.9 | 1223.5 | 307.1 | 79.0 | 46.0 |
| Additional scaling (PT) | +RefCOCO(50.6K) | 58.6 | 42.3 | 1257.4 | 314.3 | 79.8 | 47.1 |
| | +AOKVQA (66.2K) | 59.6 | 43.1 | 1288.1 | 321.4 | 80.4 | 47.5 |
| Additional scaling (SFT) | +ShareGPT4o (57.3K) | 60.2 | 43.7 | 1304.5 | 327.7 | 80.9 | 47.8 |
| | +ALLaVA Inst (70K) | 60.6 | 44.0 | 1316.2 | 331.0 | 81.3 | 48.2 |

## 5.2 IMPACT OF THE BPE IMAGE TOKENIZER

**Importance of Two-Stage Training:** Our results clearly demonstrate the necessity of the two-stage training process. Models trained only with SFT show notably lower performance compared to those that underwent both PT and SFT. This finding underscores the importance of the pretraining stage in guiding the expanded token embedding layers to learn meaningful visual representations.

**Freezing *vs*. Full Pretraining:** We observe a consistent performance advantage when freezing the text token embeddings during pretraining, *i.e.*, PT(freeze), compared to training all embeddings, *i.e.*, PT(full). For instance, in the 'LLM+VQ+BPE' setting, PT(freeze)+SFT outperforms PT(full)+SFT across all benchmarks, with notable improvements in $MME^p$ (1223.5 *vs*. 1144.6) and $MME^c$ (307.1 *vs*. 284.3). This suggests that allowing the text embeddings to update during pretraining may interfere with the model's ability to focus on learning visual representations.

**Performance Gains:** The integration of our BPE Image Tokenizer improves model performance across all benchmarks, as evidenced by comparing the 'LLM+VQ' and 'LLM+VQ+BPE' rows in Table 5.1. These gains indicate that the BPE Image Tokenizer provides our `Being-VL-0` model with better multimodal understanding. The improvements is consistent across different training methods, also highlighting the robustness of our approach.

## 5.3 IMPACT OF ADDITIONAL DATA SCALING

To investigate the scalability of our approach, we incrementally add more datasets to both the pretraining and SFT phases. The results, shown in the lower part of Table 5.1, reveal a clear trend of performance improvement with increased data volume.

**Pretraining Data Scaling:** Adding RefCOCO (50.6K) and AOKVQA (66.2K) to the pretraining phase leads to consistent improvements across all benchmarks. For instance, VQAv2 scores increase from 57.1 to 59.6, and MMBench scores rise from 40.9 to 43.1. This suggests that our model benefits from exposure to a wider range of visual concepts and question-answering patterns during pretraining.

**SFT Data Scaling:** Further improvements are observed when adding ShareGPT4o (57.3K) and ALLaVA Inst (70K) to the SFT phase. The consistent improvement across different benchmarks indicates that our approach effectively leverages additional data to enhance multimodal understanding.

**Scalability Potential:** The continuous performance improvements with data scaling suggest that our training paradigm has not yet reached its upper limit. This finding is encouraging, as it implies that further performance gains could be achieved with larger datasets or more diverse data sources.

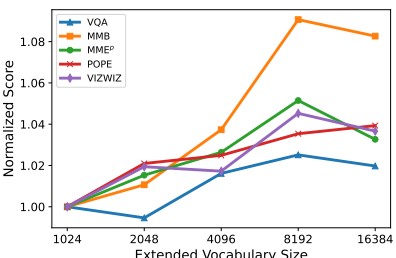
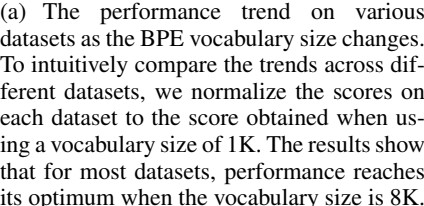
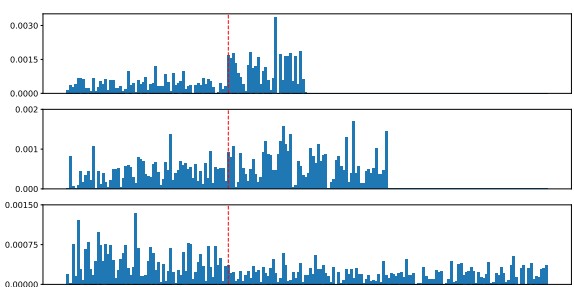

(a) The performance trend on various datasets as the BPE vocabulary size changes. To intuitively compare the trends across different datasets, we normalize the scores on each dataset to the score obtained when using a vocabulary size of 1K. The results show that for most datasets, performance reaches its optimum when the vocabulary size is 8K.

(b) Visualization of token embedding weights with different vocabulary sizes. From top to bottom, they respectively represent the 4K/8K/16K versions. The vertical axis represents the mean absolute value of weights across all layers. The horizontal axis represents the range of corresponding image token IDs. The left side of the red line represents the range of VQ-GAN's codebook (8K). The right area represents the range of correspondingly expanded BPE vocabulary (4K/8K/16K).

Figure 3: (Left) The relationship between model performance and the size of the BPE vocabulary. (Right) The visualization of model weights for tokens usage under different vocabularies.

## 5.4 IMPACT OF BPE VOCABULARY AND TOKEN USAGE PATTERNS

We conduct a detailed analysis of how the vocabulary of the BPE Image Tokenizer affects model performance and token usage patterns. Figure 3(a) illustrates the relationship between vocabulary size and normalized performance scores across different benchmarks. We observed that when the vocabulary size is lower than 8K (which equals the codebook size of the VQ-GAN model), the performance improves with the increase in vocabulary size. However, when the vocabulary size reaches 16K, all models on datasets except POPE show performance decline. This trend suggests an optimal vocabulary size should balances efficiency and model complexity.

To further verify this finding and gain deeper insights into how models utilize different vocabularies, we visualize the token embedding weights of models trained with different vocabulary sizes. Figure 3(b) shows the absolute values of these weights, averaged across all layers. We observe three distinct patterns: 1) With a smaller extended vocabulary (4K), the model tends to utilize more extended tokens from the BPE image tokenizer; 2) While with a larger extended vocabulary (16K), the model uses more of the original token IDs covered by the VQ-GAN's codebook; 3) Only when using a balanced extended vocabulary size (8k) does the model's token selection become more uniform, achieving the best performance.

We hypothesize that this phenomenon occurs because the original token IDs covered by the VQ-GAN codebook contain fine-grained visual information, while the combined token IDs provided by the BPE Image Tokenizer capture higher-level semantic concepts. A balanced utilization of these two types of information appears to be most conducive to the model's overall learning and performance. These findings may provide valuable insights for future MLLM training design.

## 6 RELATED WORK

Recent advancements in Multimodal Large Language Models (MLLMs) have demonstrated remarkable capabilities in various tasks, including visual question answering, image captioning, and cross-modal retrieval (Yin et al., 2023). The evolution of MLLMs can be broadly categorized into two main approaches: late-fusion models with specialized encoders and early-fusion token-based models.

Traditional late-fusion MLLMs typically employ distinct specialized designs for different modalities (Team, 2024). In this approach, modality-specific modules are trained separately using specialized data, followed by additional alignment steps to achieve late-fusion of different modalities. For instance, in the image modality, CLIP-based visual encoders (Radford et al., 2021; Fang et al., 2023)

are first pre-trained on extensive image-caption datasets, then aligned with text-only LLM backbones (Touvron et al., 2023; Peng et al., 2023) using additional data. Notable examples of this approach include Flamingo (Alayrac et al., 2022), LLaVA (Liu et al., 2024b;a), IDEFICS (Laurençon et al., 2024), and Emu (Sun et al., 2023). However, these models often face challenges in modality alignment, leading to issues such as hallucination in MLLMs (Bai et al., 2024).

To address these challenges, recent research has explored early-fusion approaches through unified representations, proposing token-based methods for multimodal learning (Team, 2024; Lu et al., 2024; Zheng et al., 2024). These methods typically utilize Vector Quantization (VQ) models (Van Den Oord et al., 2017; Razavi et al., 2019; Esser et al., 2021) to convert images into discrete tokens. The concept of token-based multimodal learning was initially explored in studies such as BEiT (Bao et al., 2021), which introduced a self-supervised method for acquiring visual representations based on tokenized image patches. This idea was further developed in works like Cm3 (Aghajanyan et al., 2022) and CM3Leon (Yu et al., 2023), which enabled joint reasoning across modalities and scaled up to autoregressive text-to-image generation. More recent models like Gemini (Team et al., 2023), Unicode (Zheng et al., 2024), and Chameleon (Team, 2024) have adopted end-to-end token-based approaches for multimodal learning.

While these token-based MLLMs demonstrate enhanced reasoning and generation capabilities across various modalities without requiring modality-specific components, they still face challenges in representation learning and alignment (Baltrušaitis et al., 2018). Our proposed BPE Image Tokenizer paradigm addresses these challenges by adhering more closely to the learning method of text-only LLMs. Unlike current token-based approaches, our method directly incorporates crucial structural prior information into the tokens through explicit merging. This approach enables Transformer models to learn input data more effectively, as supported by both our theoretical analysis and experimental results.

The key distinction of our work lies in its focus on optimizing the tokenization process itself, rather than relying solely on pre-trained visual encoders or simple quantization. While traditional visual encoders can achieve high compression rates, their encoded embeddings often depend on specialized decoders for interpretation. In contrast, our BPE Image Tokenizer creates tokens that are directly meaningful to the Transformer model, facilitating more effective learning of visual information without the need for specialized decoders. This approach bridges the gap between visual and textual modalities more seamlessly, potentially leading to more robust and versatile MLLMs.

# 7 CONCLUSIONS, LIMITATIONS AND FUTURE WORK

In this paper, we proposed a novel paradigm for MLLM training, demonstrating improvements in image understanding capabilities across various benchmarks, even with limited training data. Our approach centers on the novel BPE Image Tokenizer, which effectively combines image token IDs to facilitate better integration of visual information into MLLMs. We provided theoretical insights into the importance of tokenization for learning 2D sequence data with Transformer models, supporting our empirical findings and offering a new perspective on processing visual information in language models. Our traind `Being-VL-0` model not only showcased the effectiveness of the BPE Image Tokenizer but also demonstrated the scalability of our approach.

While our results are promising, we acknowledge several limitations of our current work. The experiments were conducted with limited computational resources and training data compared to state-of-the-art MLLMs, potentially understating the full potential of our approach. Additionally, the simplified 2D Markov process used in our theoretical analysis, while instructive, may not fully capture the complexities of real-world image data.

Based on the findings and limitations, we propose several directions for future work. Scaling up the training data and model size would allow us to fully explore the potential of our BPE Image Tokenizer approach in large-scale MLLMs. Investigating the applicability of our method to other visual tasks, including video understanding, could significantly broaden its impact. Exploring more sophisticated tokenization strategies that can better capture the complex dependencies in real-world multimodal data is another promising avenue.

ACKNOWLEDGMENTS

This work was supported by NSFC under Grant 62450001 and 62476008. The authors would like to thank the anonymous reviewers for their valuable comments and advice.

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

# A  THEORETICAL DISCUSSIONS

## A.1  INFORMATION LOSS OF THE TOKENIZER

Let $\boldsymbol{X}$ denote the original image, $\boldsymbol{V}$ denote the VQ-tokenized sequence, and $\boldsymbol{T}$ denote the final BPE-tokenized sequence. The information loss introduced by the BPE tokenization process can be quantified using conditional entropy:

$$L_{bpe} = H(\boldsymbol{X}|\boldsymbol{T}) - H(\boldsymbol{X}|\boldsymbol{V}). \tag{6}$$

This quantity represents the increase in uncertainty about the original image $\boldsymbol{X}$ when using BPE tokens $\boldsymbol{T}$ compared to using VQ tokens $\boldsymbol{V}$. Since

$$L_{bpe} = H(\boldsymbol{X}|\boldsymbol{T}) - H(\boldsymbol{X}|\boldsymbol{V}) = [H(\boldsymbol{X},\boldsymbol{T}) - H(\boldsymbol{T})] - [H(\boldsymbol{X},\boldsymbol{V}) - H(\boldsymbol{V})], \tag{7}$$

and recall that mutual information can be expressed as:

$$\begin{aligned} I(\boldsymbol{X};\boldsymbol{V}) &= H(\boldsymbol{X}) + H(\boldsymbol{V}) - H(\boldsymbol{X},\boldsymbol{V}), \\ I(\boldsymbol{X};\boldsymbol{T}) &= H(\boldsymbol{X}) + H(\boldsymbol{T}) - H(\boldsymbol{X},\boldsymbol{T}). \end{aligned} \tag{8}$$

Substituting back into equation 7 and we get

$$\begin{aligned} L_{bpe} &= [H(\boldsymbol{X}) + H(\boldsymbol{T}) - I(\boldsymbol{X};\boldsymbol{T}) - H(\boldsymbol{T})] - [H(\boldsymbol{X}) + H(\boldsymbol{V}) - I(\boldsymbol{X};\boldsymbol{V}) - H(\boldsymbol{V})] \\ &= [H(\boldsymbol{X}) - I(\boldsymbol{X};\boldsymbol{T})] - [H(\boldsymbol{X}) - I(\boldsymbol{X};\boldsymbol{V})] \\ &= I(\boldsymbol{X};\boldsymbol{V}) - I(\boldsymbol{X};\boldsymbol{T}). \end{aligned} \tag{9}$$

Therefore, the information loss can be expressed as the reduction in mutual information between the original image and its tokenized representation. This formulation is particularly useful as it directly quantifies how much information about the original image is preserved through the tokenization process.

Next, we need to analyze how each BPE merge operation affects the mutual information $I(\boldsymbol{X};\boldsymbol{T})$. For any given BPE token sequence $\boldsymbol{T}$, we can decompose $H(\boldsymbol{X}|\boldsymbol{T})$ based on individual tokens:

$$H(\boldsymbol{X}|\boldsymbol{T}) = \sum_t P(t) H(\boldsymbol{X}|\boldsymbol{T} = t). \tag{10}$$

During a merge operation that combines tokens $(t_i, t_j)$ into $t_m$, the change in conditional entropy can be expressed as:

$$\Delta H = P(t_m) H(\boldsymbol{X}|\boldsymbol{T} = t_m) - [P(t_i, t_j) H(\boldsymbol{X}|\boldsymbol{T} = t_i, t_j)]. \tag{11}$$

For the whole token distribution, the increase in conditional entropy can be quantified using the KL divergence:

$$H(\boldsymbol{X}|t_m) = H(\boldsymbol{X}|t_i, t_j) + \text{KL}(P(\boldsymbol{X}|t_i, t_j) || P(\boldsymbol{X}|t_m)), \tag{12}$$

where $\text{KL}(\cdot)$ represents the KL divergence. In this equation,

- $H(\boldsymbol{X}|t_i, t_j)$ represents the original uncertainty about $\boldsymbol{X}$ given the separate tokens
- $\text{KL}(P(\boldsymbol{X}|t_i, t_j) || P(\boldsymbol{X}|t_m))$ represents the extra uncertainty introduced by merging the tokens
- The equation shows that no other sources of information loss exist beyond these two terms.

This leads to an initial upper bound on the total information loss:

$$L_{bpe} \leq \sum_{merges} \text{KL}(P(\boldsymbol{X}|t_i, t_j) || P(\boldsymbol{X}|t_m)). \tag{13}$$

The summation here represents the sum over all merges during the BPE process. Considering that BPE performs merges based on frequency patterns and with minimum merge frequency threshold $p_{\min}$, we can relax the inequality to get a more interpretable bound:

$$L_{bpe} \leq (|D_{bpe}| - |D_{vq}|) \times (-p_{\min} \log(p_{\min})). \tag{14}$$

Here, $|D_{vq}|$ is the size of the VQ codebook, and $|D_{bpe}|$ is the size of the vocabulary after BPE extension.

To put this bound in perspective, consider a typical configuration:

- $|D_{vq}| = 8192$ (VQ codebook size)
- $|D_{bpe}| = 8192 + 8192$ (vocabulary size after BPE extension)
- $p_{\min} = 0.01$ (minimum merge frequency)

The upper bound on information loss for the whole vocabulary would be:

$$L_{bpe} \leq (8192 + 8192 - 8192) \times (-0.01 \times \log(0.01)) \approx 377.3 \text{ bits.} \tag{15}$$

For a single image, the original VQ tokens ($32 \times 32$ patches) contain $32 \times 32 \times \log_2 8192 = 13312$ bits information, and the per token loss of the extended BPE vocabulary is $L_{bpe}/(|D_{bpe}| - |D_{vq}|) \approx 0.046$ bits. We can calculate that the max loss ratio of the single image is only $32 \times 32 \times 0.046/13312 \approx 0.35\%$, which is a relatively small information loss. Considering the benefits brought by BPE tokenization as discussed earlier, we believe this loss is acceptable.

## A.2 FULL PROOFS

*Proof of Proposition 1.* We begin by proving equation 3. For any unigram model $Q \in \mathcal{Q}_{\text{1-gram}}$ on the tokens, there exist distributions $Q_\#$ and $Q_{\text{tok}}$, supported on $\mathbb{N}$ and Dict (dictionary of the tokenizer) respectively, such that for all token sequence $\boldsymbol{t} = (t_1, ..., t_{|\boldsymbol{t}|})$,

$$Q(\boldsymbol{t}) = Q_\#(|\boldsymbol{t}|) \prod_{r=1}^{|\boldsymbol{t}|} Q_{\text{tok}}(t_r).$$

Consequently, for every $m \in \mathbb{N}$ we have

$$\mathcal{L}_m(Q) = -\frac{1}{m^2} \mathbb{E}\left[\log Q_\#(|\operatorname{flat}(\boldsymbol{X})|)\right] - \frac{1}{m^2} \mathbb{E}\left[\sum_{t \in \operatorname{flat}(\boldsymbol{X})} \log Q_{\text{tok}}(t)\right]$$

$$\overset{(i)}{=} -\frac{1}{m^2} \mathbb{E}\left[\log Q_\#(m^2)\right] - \frac{1}{m^2} \sum_{i=1}^{m} \sum_{j=1}^{m} \mathbb{E}\left[\log Q_{\text{tok}}(X_{ij})\right]$$

$$\geq -\frac{1}{m^2} \sum_{i=1}^{m} \sum_{j=1}^{m} \mathbb{E}\left[\log Q_{\text{tok}}(X_{ij})\right]$$

$$\overset{(ii)}{=} -\sum_{a \in \mathcal{C}} \pi(a) \log(Q_{\text{tok}}(a))$$

$$\geq H(\pi),$$

where in $(i)$ we apply the definition of $\operatorname{flat}(\cdot)$, and in $(ii)$ we use the fact that $\pi$ is the stationary distribution for each column/row as assumed in Scenario 1/Scenario 2. Thus, equation 3 is proved.

Next, we proceed to prove equation 4. For an arbitrary model $Q$, we have

$$m^2 \mathcal{L}_m(Q) = -\mathbb{E}[\log(Q(\operatorname{flat}(\boldsymbol{X})))]$$
$$= \mathbb{E}[\log(P^*(\operatorname{flat}(\boldsymbol{X}))/Q(\operatorname{flat}(\boldsymbol{X})))] - \mathbb{E}[\log(P_m^*(\operatorname{flat}(\boldsymbol{X}))]$$
$$= D_{\text{KL}}(P_m^* || Q) + H(P_m^*),$$
$$\geq H(P_m^*).$$

where $P_m^*$ denotes the actual joint distribution of $\mathrm{flat}(\boldsymbol{X})$ and $D_{\mathrm{KL}}(P_m^* \| Q)$ denotes the KL divergence between $P_m^*$ and $Q$. Under Scenario 1, leveraging the independence between columns we derive

$$H(P_m^*)$$

$$= -\sum_{j=1}^{m} \left\{ \mathbb{E}[\log \pi(X_{1j})] + \sum_{i=2}^{m} \mathbb{E}[\log P(X_{ij} \mid X_{i-1,j})] \right\}$$

$$= -\sum_{j=1}^{m} \left\{ \sum_{a \in \mathcal{C}} \pi(a) \log(\pi(a)) + (m-1) \sum_{a \in \mathcal{C}} \sum_{a' \in \mathcal{C}} \pi(a) P(a' \mid a) \log\left(P(a' \mid a)\right) \right\}$$

$$= m H(\pi) + m(m-1) H_{\infty}.$$

This result is also applicable under Scenario 2 by switching the roles of $i$ and $j$. As a result, in both scenarios we have

$$\liminf_{m \to \infty} \min_{Q} \mathcal{L}_m(Q) \geq \lim_{m \to \infty} \frac{1}{m^2} H(P_m^*) = H_{\infty}.$$

Conversely, it is evident that

$$\min_{Q} \mathcal{L}_m(Q) \leq \mathcal{L}_m(P_m^*) = \frac{1}{m^2} H(P_m^*).$$

Hence,

$$\limsup_{m \to \infty} \min_{Q} \mathcal{L}_m(Q) \leq \lim_{m \to \infty} \mathcal{L}_m(P_m^*) = H_{\infty}.$$

This finishes the proof of equation 4. $\qquad\square$

*Proof of Proposition 2.* In the one-dimensional case, where the data is a string generated from an ergodic Markov process, authors in (Rajaraman et al., 2024) have proved the following result:

**Lemma A.1** (Theorem 4.1 in (Rajaraman et al., 2024)). *Suppose a string $\boldsymbol{s}$ of length $m$ is generated from an ergodic data source supported on a finite set $\mathcal{A}$, with a transition kernel $P(\cdot \mid \cdot)$ and a stationary distribution $\pi$. Assume the transition kernel satisfies $\min_{a,a' \in \mathcal{A}} P(a' \mid a) \overset{\Delta}{=} \delta > 0$. Then, there exists a tokenizer with a dictionary containing at most $D$ tokens, along with an encoding function $\mathrm{enc}(\cdot)$ applied to $\boldsymbol{s}$, such that*

$$\limsup_{m \to \infty} \min_{Q \in \mathcal{Q}_{1\text{-gram}}} \frac{1}{m} \mathbb{E}[\log(1/Q(\mathrm{enc}(\boldsymbol{s})))])$$

$$\leq \frac{1}{1-\varepsilon} \sum_{a \in \mathcal{A}} \sum_{a' \in \mathcal{A}} \pi(a) P(a' \mid a) \log\left(1/P(a' \mid a)\right), \qquad (16)$$

*where $\varepsilon = \log(1/\delta)/(0.99 \log(D))$ and $D \in \mathbb{N}$ is an arbitrary constant that is sufficiently large.*

Building on the results of Lemma A.1, we extend these findings to images generated under Scenario 1 or 2.

In both scenarios, the generated image is composed of $m$ parallel Markov sequences, each of length $m$, sharing the same transition kernel $P(\cdot \mid \cdot)$ and stationary distribution $\pi$.

Consider Scenario 1. According to Lemma A.1 and equation 16, there exists a tokenizer equipped with a dictionary, denoted as $\mathrm{Dict}$ where $|\mathrm{Dict}| \leq D$, and an encoding function, denoted as $\mathrm{enc}_{\mathrm{col}}(\cdot)$, and there also exists a unigram model $Q_m \in \mathcal{Q}_{1\text{-gram}}$ for each $m \in \mathbb{N}$, such that

$$\limsup_{m \to \infty} \frac{1}{m} \mathbb{E}[\log(1/Q_m(\mathrm{enc}_{\mathrm{col}}(\boldsymbol{X}_{:j})))])$$

$$\leq \frac{1}{1-\varepsilon} \sum_{a \in \mathcal{C}} \sum_{a' \in \mathcal{C}} \pi(a) P(a' \mid a) \log\left(1/P(a' \mid a)\right)$$

$$= \frac{1}{1-\varepsilon} H_{\infty}.$$

Here $\boldsymbol{X}_{:j}$ represents the $j$-th column of image $\boldsymbol{X}$ and $j \in \{1, ..., m\}$ is arbitrary. For the unigram model $Q_m$, there exist distributions $Q_m^{\#}$ and $Q_m^{\text{tok}}$, supported on $\mathbb{N}$ and Dict respectively, such that for all token sequence $\boldsymbol{t} = (t_1, ..., t_{|\boldsymbol{t}|})$,

$$Q_m(\boldsymbol{t}) = Q_m^{\#}(|\boldsymbol{t}|) \prod_{r=1}^{|\boldsymbol{t}|} Q_m^{\text{tok}}(t_r).$$

Consequently, it is straightforward to derive:

$$\limsup_{m \to \infty} \frac{1}{m} \mathbb{E} \left[ \sum_{t \in \text{enc}_{\text{col}}(\boldsymbol{X}_{:j})} \log(1/Q_m^{\text{tok}}(t)) \right] \leq \frac{1}{1-\varepsilon} H_\infty. \tag{17}$$

To construct a qualified tokenizer for image $\boldsymbol{X}$, we can simply use the dictionary Dict, and define the encoding function as

$$\text{enc}(\boldsymbol{X}) = (\text{enc}_{\text{col}}(\boldsymbol{X}_{:1}), ..., \text{enc}_{\text{col}}(\boldsymbol{X}_{:m})),$$

which concatenates the tokens returned by each column. Subsequently, let $\tilde{Q}_m \in \mathcal{Q}_{\text{1-gram}}$ be the unigram model defined by

$$\tilde{Q}_m(\boldsymbol{t}) = Q_m^{\text{unif}}(|\boldsymbol{t}|) \prod_{r=1}^{|\boldsymbol{t}|} Q_m^{\text{tok}}(t_r).$$

Here $Q_m^{\text{unif}}$ is the uniform distribution over $\{1, 2, ..., m^2\}$, and $Q_m^{\text{tok}}$ is as previously defined. It then follows that:

$$\mathcal{L}_m(\tilde{Q}_m \circ \text{enc})$$

$$= \frac{1}{m^2} \mathbb{E}[\log(1/Q_m^{\text{unif}}(|\text{enc}(\boldsymbol{X})|))] + \frac{1}{m^2} \mathbb{E} \left[ \sum_{j=1}^{m} \sum_{t \in \text{enc}_{\text{col}}(\boldsymbol{X}_{:j})} \log(1/Q_m^{\text{tok}}(t)) \right]$$

$$= \frac{2 \log(m)}{m^2} + \frac{1}{m} \mathbb{E} \left[ \sum_{t \in \text{enc}_{\text{col}}(\boldsymbol{X}_{:j})} \log(1/Q_m^{\text{tok}}(t)) \right]$$

Finally, by equation 17

$$\limsup_{m \to \infty} \min_{Q \in \mathcal{Q}_{\text{1-gram}}} \mathcal{L}_m(Q \circ \text{enc}) \leq \limsup_{m \to \infty} \mathcal{L}_m(\tilde{Q}_m \circ \text{enc}) \leq \frac{1}{1-\varepsilon} H_\infty,$$

thereby proving equation 5.

The proof of Scenario 2 is quite similar and is omitted here. $\qquad\square$

## B  IMPLEMENTATION DETAILS

### B.1  ADDITIONAL PSEUDO CODES

The additional functions used in Algorithm 4.1 are shown in Algorithm B.1 and B.2.

---

**Algorithm B.1** Update Adjacency Matrix

---

1: **procedure** UPDATEMATRIX($D$)
2:  $A \leftarrow \text{zeros}(v, v)$
3:  **for** each $d \in D$ **do**
4:    **if** $d$ is 1D **then**
5:      **for** $j \leftarrow 1$ to $|d| - 1$ **do**
6:        $A[d_j, d_{j+1}] \leftarrow A[d_j, d_{j+1}] + 1$
7:      **end for**
8:    **else**
9:      **for** each $(x, y)$ in $d$ **do**
10:        Update $A$ for neighbors of $(x, y)$
11:      **end for**
12:    **end if**
13:  **end for**
14:  **return** $A$
15: **end procedure**

---

**Algorithm B.2** Find Most Frequent Pair

---

1: **procedure** MAXFREQPAIR($A$)
2:  $U \leftarrow \text{UpperTri}(A)$
3:  **if** $\sum U = 0$ **then**
4:    **return** $(null, 0)$
5:  **end if**
6:  $(i, j) \leftarrow \arg\max U$
7:  **return** $((i, j), U_{ij})$
8: **end procedure**

---

### B.2  HYPERPARAMETERS FOR FIGURE 2(B) AND 2(C)

In the experiment shown in Figure 2(b), we experimented all parameter combinations listed in Table B.1 for the Transformer model. For the experiment in Figure 2(c) that used a BPE tokenizer, except for the layers shown, all other parameters used the minimum values from Table B.1.

Table B.1: Hyperparameters for Transformer models

| Hyperparameter | Value / Range |
|---|---|
| BPE dictionary size | 10 |
| batch size | {8, 16, 32} |
| gradient acc. steps | 1 |
| learning rate | 2e-3 |
| weight decay | 1e-3 |
| scheduler | cosine |
| optimizer | adamw |
| dropout | 0 |
| number of heads | {1, 2, 4, 8, 16} |
| number of layers | {2, 4, 6, 8} |
| embedding dimension | {10, 20, 30, 40} |

### B.3 HYPERPARAMETERS FOR THE VQ-GAN MODEL

The hyperparameters for the VQ-GAN model used in our experiments are shown in Table B.2.

Table B.2: Hyperparameters for the VQ-GAN model

| Hyperparameter | Value |
|---|---|
| embedding dimension | 256 |
| codebook size | 8192 |
| z_channels | 256 |
| resolution | 512 |
| dropout | 0 |

### B.4 HYPERPARAMETERS FOR TRAINING MLLM

The hyperparameters for pre-training / supervised-fine-tuning MLLM are shown in Table B.3.

Table B.3: Hyperparameters for training MLLM

| Hyperparameter | PT | SFT |
|---|---|---|
| batch size | 1 | 1 |
| gradient accumulation | 2 | 4 |
| learning rate | 1e-3 | 3e-5 |
| learning schedule | cosine | cosine |
| warmup ratio | 0.03 | 0.03 |
| weight decay | 0 | 0 |
| epoch | 3 | 2 |
| optimizer | AdamW | AdamW |
| deepspeed stage | 2 | 2 |

### B.5 DETAILS OF THE PIPELINE

In this section, we provide more detailed descriptions of our pipelines as below.

#### B.5.1 LLM+VQ+BPE PIPELINE

Our complete pipeline (LLM+VQ+BPE) consists of three main components:

- Image Quantization: We employ a pretrained VQ-GAN model (with a codebook size of 8192) to quantize input images into a sequence of token IDs. This process converts images of any size into a 1024-dimensional ID tensor.

- BPE Image Tokenizer: This component, which distinguishes our approach from previous methods, processes the quantized token IDs using our trained BPE tokenizer. The tokenizer learns to merge frequently co-occurring tokens, creating a new vocabulary that captures higher-level visual patterns.

- Multimodal Learning: The processed image token IDs are combined with text token IDs and fed into the LLM (Llama-3.1-8B in our experiments). Before training, we expand the token embedding layers of the base model to accommodate both the VQ-GAN token IDs (8192) and the BPE vocabulary (e.g., 4096), along with special tokens for marking image positions.

### B.5.2 LLM+VQ Baseline

The LLM+VQ variant serves as our baseline, following a simpler approach:

- It uses the same VQ-GAN model for image quantization as in LLM+VQ+BPE.
- The quantized token IDs are directly combined with text token IDs, bypassing the BPE tokenizer stage.
- The token embedding layers are only expanded to accommodate the VQ-GAN token IDs and special tokens.

### B.5.3 Training Process

Both variants undergo the same two-stage training process:

- Image Understanding Pretraining (PT): During this stage, we train the expanded token embeddings using image caption data. In the PT(freeze) setting, we freeze the original text token embeddings, while in PT(full), all embeddings are updated.
- Supervised Fine-Tuning (SFT): All parameters are unfrozen and fine-tuned using SFT data.

## C  Additional Results

### C.1  Additional Training with Llama2 version / freezing during SFT

To further investigate the effects of different base models and varying component freezing strategies, we conducted additional experiments. The results are shown in Table C.1

Table C.1: Full results with additional experiments. The red parts indicate using Llama2 as the base LLM for training. The blue parts indicate freezing the embedding corresponding to the visual token during the SFT stage.

|  | Training type | VQAv2 | MMBench | $\text{MME}^P$ | $\text{MME}^c$ | POPE | VizWiz |
|---|---|---|---|---|---|---|---|
| Llama3.1+VQ | SFT | 51.1 | 35.9 | 972.3 | 231.8 | 73.8 | 43.1 |
|  | PT(full)+SFT | 53.7 | 37.0 | 1037.2 | 261.4 | 75.3 | 44.2 |
|  | PT(freeze text)+SFT | 55.4 | 37.6 | 1054.5 | 277.0 | 76.0 | 45.3 |
| Llama2+VQ | PT(freeze text)+SFT | 54.0 | 35.7 | 991.9 | 254.2 | 75.1 | 44.4 |
| Llama3.1+VQ+BPE | SFT | 52.2 | 35.4 | 1029.7 | 269.6 | 76.3 | 45.3 |
|  | PT(full)+SFT(freeze visual) | 31.7 | 17.8 | 624.1 | 171.9 | 46.4 | 29.5 |
|  | PT(full)+SFT | 56.5 | 38.6 | 1144.6 | 284.3 | 77.3 | 45.8 |
|  | PT(freeze text)+SFT(freeze visual) | 22.5 | 13.3 | 488.7 | 143.6 | 35.2 | 21.5 |
|  | PT(freeze text)+SFT | 57.1 | 40.9 | 1223.5 | 307.1 | 79.0 | 46.0 |
| Llama2+VQ+BPE | PT(freeze text)+SFT | 56.5 | 38.1 | 1112.2 | 277.8 | 77.9 | 44.9 |
| Additional scaling (PT) | +RefCOCO(50.6K) | 58.6 | 42.3 | 1257.4 | 314.3 | 79.8 | 47.1 |
|  | +AOKVQA (66.2K) | 59.6 | 43.1 | 1288.1 | 321.4 | 80.4 | 47.5 |
| Additional scaling (SFT) | +ShareGPT4o (57.3K) | 60.2 | 43.7 | 1304.5 | 327.7 | 80.9 | 47.8 |
|  | +ALLaVA Inst (70K) | 60.6 | 44.0 | 1316.2 | 331.0 | 81.3 | 48.2 |

## C.2 ADDITIONAL EVALUATION

To assess our method's effectiveness across various capabilities, we performed comprehensive evaluations analyzing performance across distinct sub-categories. The results are shown in Table C.2 and C.3

Table C.2: Comparison of LLM+VQ+BPE and LLM+VQ performance across different categories in MME benchmark.

|  | Category | LLM+VQ+BPE | LLM+VQ |
|---|---|---|---|
| Perception | Existence | **145.00** | 113.33 |
|  | Count | **120.00** | 110.00 |
|  | Position | **106.67** | 103.33 |
|  | Color | **148.33** | 120.00 |
|  | Posters | **136.24** | 121.08 |
|  | Celebrity | **111.76** | 89.51 |
|  | Scene | **125.00** | 101.75 |
|  | Landmark | **130.25** | 110.00 |
|  | Artwork | **112.75** | 90.50 |
|  | OCR | 87.50 | **95.00** |
| Cognition | Commonsense Reasoning | **107.14** | 89.57 |
|  | Numerical Calculation | **62.50** | 50.00 |
|  | Text Translation | 95.00 | **102.50** |
|  | Code Reasoning | **42.50** | 35.00 |

Table C.3: Comparison of LLM+VQ+BPE and LLM+VQ performance across different categories in MLLM-bench. The numbers in the table represent the quantity of answers judged to be better for each respective model. "Tie" indicates the number of answers where there is no significant difference between the two models' answers.

| Category | LLM+VQ+BPE | LLM+VQ | Tie |
|---|---|---|---|
| perception | 26 | **34** | 10 |
| understanding | **52** | 33 | 25 |
| Applying | **27** | 14 | 19 |
| Analyzing | **49** | 40 | 31 |
| Evaluation | 12 | **19** | 9 |
| Creation | 7 | **9** | 4 |
| Total | **173** | 149 | 98 |

Table C.4: Evaluation results on the Nocaps and Flickr30k benchmarks. For Nocaps, we chose the validation split. For Flickr30k, we selected a 1k-image split. We compared the impact of adding BPE on the performance of image captioning tasks that require detailed understanding. Additionally, we investigated versions of LLM+VQ+BPE (*+SFT*), which used additional scaling as described earlier, to observe how more instruction tuning affects performance in image captioning tasks.

| Model | Nocaps | | | | Flickr30k | | | |
|---|---|---|---|---|---|---|---|---|
|  | CIDEr | SPICE | METEOR | ROUGE-L | CIDEr | SPICE | METEOR | ROUGE-L |
| LLM+VQ | 76.4 | 17.2 | 24.4 | 51.3 | 76.4 | 16.2 | 24.4 | 51.3 |
| LLM+VQ+BPE | 75.7 | 16.9 | 24.1 | 50.8 | 75.7 | 15.9 | 24.1 | 50.8 |
| LLM+VQ+BPE (*+SFT*) | 80.7 | 18.2 | 25.0 | 52.6 | 80.7 | 17.2 | 25.0 | 52.6 |

# D    ADDITIONAL DISCUSSION ABOUT FURTHER SCALING

Our methods with BPE Image Tokenizer represent a departure from conventional CLIP-based visual encoding methods, introducing both challenges and opportunities that warrant careful discussion.

A critical aspect of evaluating our approach is understanding the implicit data scale disparity between our method and existing CLIP-based MLLMs. Traditional visual encoders typically leverage massive pretraining datasets—CLIP's original implementation utilized 400M image-text pairs (Radford et al., 2021), while subsequent models expanded to datasets like LAION-2B, processing billions of samples across multiple epochs. In contrast, our BPE Image Tokenizer achieves competitive performance using approximately 2.78M images, representing roughly 0.1% of the training data used by encoders like CLIP-ViT-L/14.

This substantial difference in pre-training data requirements highlights a fundamental trade-off in MLLM development: while CLIP-based approaches benefit from extensive pre-training, our method demonstrates remarkable efficiency in learning visual representations from limited data. The comparable performance achieved with significantly fewer resources suggests that our approach may offer a more scalable path forward for MLLM development, particularly in scenarios where massive datasets or computational resources are not available.

While our current implementation may not achieve state-of-the-art performance compared to heavily optimized CLIP-based MLLMs, this limitation should be viewed within the broader context of our research objectives. The primary contribution of this work lies not in establishing a new performance benchmark, but rather in introducing and validating a novel training paradigm for MLLMs. The theoretical foundations and experimental results presented here serve as proof-of-concept for an alternative approach to visual-language modeling.

We acknowledge that achieving state-of-the-art performance would require:

- Substantial expansion of training data
- Significant computational resources for large-scale training
- Extensive engineering optimizations

These requirements extend beyond the scope of our current investigation, which focuses on establishing the theoretical and practical viability of our approach.

The promising results obtained with limited resources suggest exciting potential for future research. We believe our work lays the groundwork for a new direction in MLLM research, demonstrating that alternative approaches to visual-language modeling can achieve competitive performance even with limited resources. As the field continues to evolve, the principles and methods introduced in this paper may contribute to the development of more efficient and scalable MLLMs. Our future work will focus on scaling up these initial findings while preserving the core benefits of our approach.

## E   EXAMPLES

Table E.1: Output examples of models trained with and without using BPE tokenizer. The text marked in red has obvious hallucinations.

| Case | LLM+VQ+BPE | LLM+VQ |
|---|---|---|
| **Q:** Give a brief description of the image. | | |
| 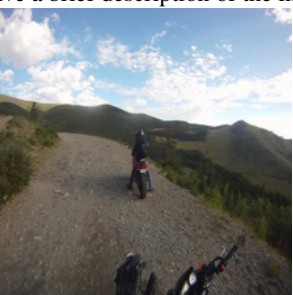 | The image appears to be taken from the perspective of someone riding a bicycle or motorcycle on a dirt road or mountain trail. We can see handlebars in the foreground, suggesting this is a first-person view from the rider. | The image features a motorcyclist riding on a dirt road, wearing a helmet and a black jacket. They are surrounded by a group of people who are taking pictures of the rider as they pass by. The motorcyclist appears to be enjoying the moment and capturing the memory. |
| **Q:** What is in the photo? | | |
| 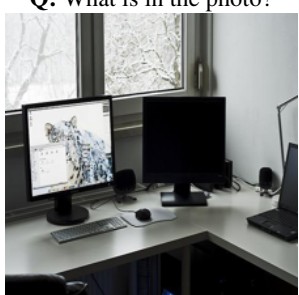 | This photo shows a workspace or home office setup. There are two computer monitors on a desk. The desk is positioned in front of a window. The desk appears to be an L-shaped or corner desk, with a clean and minimalist design. | In the photo, a desk is set up with a laptop positioned in the center, surrounded by a few other laptops and monitors. There is an empty black chair placed in front of the desk. A potted plant is located on the left side of the scene, creating a lively atmosphere. |

As shown in Table E.1, in these two cases, the version without using the BPE tokenizer, which directly uses VQ-quantized tokens for `Llama-3.1-8B` training, frequently exhibits hallucination issues. In contrast, the model trained with the BPE tokenizer can more accurately describe the information in the images, showcasing better image comprehension capabilities.

## F   BROADER IMPACT

The development of our proposed novel paradigm for training MLLMs has the potential to advance artificial intelligence and its applications across various domains, including medical imaging and autonomous systems. However, this progress also raises important ethical considerations, such as privacy concerns and the potential for bias in AI systems. As this technology evolves, it is crucial to balance innovation with responsible development, necessitating collaboration between researchers, ethicists, and policymakers. Future work should not only focus on technical improvements but also on ensuring that these advancements benefit society while mitigating potential risks. This research thus opens new avenues for AI capabilities while underscoring the importance of ethical considerations in technological progress.

## G    COMPUTING RESOURCES

We list the hardware resources used in Table G.1.

Table G.1: Computing resources for our experiments.

| CPU | GPU | RAM |
|---|---|---|
| Intel 3GHz $\times$ 64 | Nvidia A800 (80GB) $\times$ 8 | 1024GB |

## H    LICENSES

In our code, we have used the following libraries which are covered by the corresponding licenses:

- Numpy (BSD-3-Clause license)
- PyTorch (BSD-3-Clause license)
- Transformers (Apache license)
- Numba (BSD-2-Clause license)

