# OpenReview forum: "From Pixels to Tokens: Byte-Pair Encoding on Quantized Visual Modalities"
_ICLR.cc/2025/Conference — ICLR 2025 Poster_

### Official Review · Reviewer_UsVN · 2024-10-28

**Soundness:** 3
**Presentation:** 3
**Contribution:** 2
**Rating:** 8
**Confidence:** 3

**Summary:**

This paper presents a novel BPE image tokenizer that brings byte-pair encoding (BPE) to image tokenization, enhancing multimodal large language models (MLLMs) in aligning visual and textual information. By embedding structural priors into image tokens, the method shows promise in cross-modal tasks and scalability, offering a new approach to unified visual and text processing.

**Strengths:**

1.This paper creatively adapts byte-pair encoding (BPE) for images, aiming to make visual data work more seamlessly with text in multimodal models.

2.The approach integrates structural information directly into image tokens, which could help models better understand and align visuals with text, showing solid potential in cross-modal tasks.

**Weaknesses:**

1.Theoretical framework has several notable limitations:
  1.1Lack of Multimodal Fusion Analysis: The paper’s theoretical analysis is focused on 2D image data alone and does not delve into how the BPE tokenizer facilitates the fusion of visual and textual information. Multimodal tasks typically require deep semantic and structural alignment across modalities, which is not sufficiently addressed in this analysis. This omission limits the theoretical support for the tokenizer’s efficacy in a multimodal model context.
  1.2Absence of Analysis on Information Loss in Tokenization: The paper lacks a theoretical exploration of the potential information loss from BPE tokenization, such as the simplification of high-frequency visual details. There is no quantification of how the loss of these details might impact overall model performance. This gap in the analysis leaves the question of how well the BPE tokenizer preserves image details unanswered.

2.A notable limitation of this paper is its focus on evaluating the BPE image tokenizer primarily through VQA-like tasks, which generally require only broad semantic alignment across modalities. While effective for assessing general multimodal comprehension, these tasks may not fully capture the demands of applications like image segmentation or image caption, where finer-grained visual detail and spatial relationships are crucial. Without evaluation on these more intricate tasks, it remains unclear how well the method handles scenarios that require detailed visual representation, potentially limiting its applicability to real-world multimodal use cases that demand high visual fidelity.

**Questions:**

1.Applicability of the Theoretical Model: How does a simplified 2D Markov process adequately capture the complex structure of real-world image data?

2.Sensitivity to Information Loss: How is the potential impact of information loss in tokenization, especially for detail-sensitive tasks, theoretically assessed?

3.Task Representativeness and Generalization: How can results on VQA-like tasks ensure performance on precision-demanding tasks like image caption?

---

> ### Author Response · Authors · 2024-11-22
>
> We deeply appreciate the time and effort the reviewer has invested in reviewing our work, along with the valuable feedback and insightful suggestions provided. The valuable feedback and insightful suggestions are of great significance to us. We hope that the following responses adequately address the reviewer's questions and concerns.
>
> > W1.1: About the analysis of how BPE tokenizer enables visual-textual information fusion in multimodal contexts.
>
> We appreciate the reviewer's insightful comment regarding the theoretical analysis of multimodal fusion. While our theoretical framework primarily focuses on 2D image data, it establishes fundamental guarantees for the integration of our BPE image tokenizer with the transformer architecture.
>
> Proposition 2 provides a performance bound demonstrating that our BPE image tokenizer enhances the transformer's learning capabilities. Specifically, it proves that **an appropriately designed tokenizer can enable the transformer model to achieve a loss close to the optimal unconstrained loss** $H_\infty$ even under worst-case conditions. This theoretical result offers key insights into why our approach strengthens the model's multimodal understanding. Intuitively, the text-BPE inspired design also creates natural alignment between image and text tokenization strategies, facilitating more effective multimodal learning.
>
> We acknowledge the point that a more comprehensive theoretical analysis of multimodal fusion mechanisms would provide additional guarantees. However, developing such a theoretical framework presents substantial challenges, as many core aspects of transformer architectures themselves still lack theoretical understanding. Given these constraints, we adopted the common approach combining theoretical insights with extensive empirical validation. This is widely accepted in the research community for evaluating novel ideas.
>
> We value the reviewer's suggestion and agree that extending our theoretical framework to analyze the detailed interactions between image tokenization and transformer mechanisms represents a promising direction for future work. Such analysis could provide deeper insights into multimodal fusion and guide further architectural designs.
>
> > Q1: About how does the 2D Markov process capture the real-world image data?
>
> As we already discussed in Section 3: "This simplification is intuitive, as pixels in an image often exhibit conditional dependencies with other pixels at specific horizontal and vertical distances. Consequently, real-world image data can be viewed as a composite of multiple such simplified processes."
>
> More formally, our 2D Markov process is motivated by a fundamental observation about natural images: pixels typically exhibit strong dependencies with other pixels at specific horizontal and vertical distances. This intuition can be formalized as follows:
>
> For any pixel $X_{i,j}$ in a real image, its value is intuitively influenced by a combination of multiple conditional dependencies:
>
> $$
> P(X_{i,j}|X_{<i,j}) = \sum_{k=1}^K w_k P_k(X_{i,j}|X_{i-k,j}, X_{i,j-k})
> $$
>
> where $w_k$ are importance weights ($\sum_k w_k = 1$), $P_k$ represents the k-distance dependency model, K is the maximum dependency distance considered. This formulation captures several key properties of real images:
>
> - **Spatial Locality**: The strongest dependencies are typically local, reflected in larger weights $w_k$ for smaller k;
> - **Directional Structure**: By explicitly modeling horizontal and vertical dependencies, we capture the primary axes along which visual patterns typically align;
> - **Multi-scale Dependencies**: Different k values capture dependencies at different scales, from fine details to broader structures.
>
> While our primary theoretical findings are based on a single 2D k-th order Markov process, it's important to note that the bound established in Proposition 2 naturally extends to linear combinations of such models. For clarity and readability, we chose to present our proofs using the simplest case of a single 2D Markov process in the paper.

---

> ### Author Response · Authors · 2024-11-22
>
> > W1.2 & Q2: About the theoretical analysis on BPE image tokenizer's information loss
>
> Thank you for pointing this out. We agree that analyzing the information loss of the BPE image tokenizer is crucial for understanding its capability to handle fine-grained details. We have addressed this concern by adding a detailed discussion about information loss in **Section A.2** of our revision. The reviewer could check the updated PDF for the full analysis.
>
> In summary, the main conclusion of Section A.2 is that we've proven there is an upper bound on the information loss caused by BPE:
> $$
> L_{bpe} \leq (|D_{bpe}| - |D_{vq}|) \times (-p_{\min}\log(p_{\min})).
> $$
> Here, $|D_{vq}|$ is the size of the VQ codebook, $|D_{bpe}|$ is the size of the vocabulary after BPE extension, and $p_{\min}$ is the minimum merge frequency threshold. To put this bound in perspective, consider a typical configuration:
>
> - $|D_{vq}|=8192$ (VQ codebook size)
> - $|D_{bpe}|=8192+8192$  (vocabulary size after BPE extension)
> - $p_{\min} = 0.01$ (minimum merge frequency)
>
> The upper bound on information loss for the whole vocabulary would be: $L_{bpe} \leq (8192+8192-8192) \times (-0.01\times \log(0.01)) \approx 377.3 ~ \mathrm{bits}$. For a single image, the original VQ tokens ($32\times 32$ patches) contain $32 \times 32 \times \log_2 8192 = 13312 ~ \mathrm{bits}$ information, and the per token loss of the extended BPE vocabulary is $L_{bpe}/(|D_{bpe}| - |D_{vq}|) \approx 0.046~\mathrm{bits}$. We can calculate that the max loss ratio of the single image is only $32\times 32\times 0.046 / 13312\approx 0.35$%, which is a relatively small information loss. Considering the benefits brought by BPE tokenization as discussed earlier, we believe this loss is acceptable.

---

> ### Author Response · Authors · 2024-11-22
>
> > W2 & Q3: About the limitation that VQA-based evaluation may not reflect performance on precision-demanding visual tasks.
>
> We would like to emphasize that our evaluations already include tasks requiring high-precision detail comprehension. For instance, MME includes OCR/Position/Text tasks, while MMBench covers OCR, Object localization, and Attribute recognition. These tasks specifically test fine-grained perception abilities. Also, the VQA format enables fair score computation for accurate model comparison, which is why these benchmarks are widely accepted as standard evaluations for MLLMs.
>
> To better illustrate our BPE Image Tokenizer's performance on tasks requiring detailed image understanding, we present and analyze specific metrics from the MME benchmark:
>
> | Category   | Subcategory           | LLM+VQ+BPE | LLM+VQ |
> | ---------- | --------------------- | ---------- | ------ |
> | Perception | Existence             | **145.0**  | 113.33 |
> |            | Count                 | **120.0**  | 110.0  |
> |            | Position              | **106.67** | 103.33 |
> |            | Color                 | **148.33** | 120.0  |
> |            | Posters               | **136.24** | 121.08 |
> |            | Celebrity             | **111.76** | 89.51  |
> |            | Scene                 | **125.0**  | 101.75 |
> |            | Landmark              | **130.25** | 110.0  |
> |            | Artwork               | **112.75** | 90.5   |
> |            | OCR                   | 87.5       | **95.0**   |
> | Cognition  | Commonsense Reasoning | **107.14** | 89.57  |
> |            | Numerical Calculation | **62.5**   | 50.0   |
> |            | Text Translation      | 95.0       | **102.5**  |
> |            | Code Reasoning        | **42.5**   | 35.0   |
>
> While LLM+VQ+BPE shows slightly lower performance in tasks requiring detail understanding like OCR and Text Translation, the gap is not substantial. Moreover, it maintains clear advantages in most tasks, also excelling in Position and Code Reasoning tasks that require detail understanding. This suggests that the minor trade-off in fine-grained details doesn't significantly impact overall performance.
>
> We also acknowledge the reviewer's concerns about VQA-style evaluation limitations. Therefore, we conducted additional evaluations on MLLM-bench, an open-ended benchmark where responses are evaluated by GPT-4v (choosing which model answers better). This benchmark also includes precision-demanding tasks like OCR/Text Recognition/Object Recognition, etc. The results are as follows:
>
> |               | LLM+VQ+BPE | LLM+VQ | Tie |
> | ------------- | ---------: | -----: | --: |
> | perception    |         26 | **34** |  10 |
> | understanding |     **52** |     33 |  25 |
> | Applying      |     **27** |     14 |  19 |
> | Analyzing     |     **49** |     40 |  31 |
> | Evaluation    |         12 | **19** |   9 |
> | Creation      |          7 |  **9** |   4 |
> | Total         |    **173** |    149 |  98 |
>
> In this table, the numbers represent the quantity of answers judged to be better for each respective model. "Tie" indicates the number of answers where there is no significant difference between the two models' answers. The results show that while our BPE image tokenizer experiences some performance decrease in Perception and Evaluation tasks (which require more detail understanding), it demonstrates significant improvements in other tasks. Notably, for Analyzing tasks that demand both fine-grained understanding and global reasoning, our method achieves overall improvement despite the slight trade-off in detail perception.
>
> We believe that our supplementary experimental results demonstrate that although our approach may lead to a slight loss in detail understanding, this degradation is within acceptable range. Meanwhile, the performance improvements achieved across various tasks through this trade-off could bring meaningful value to the practical applicability of MLLMs.
>
> ---
>
> We thank the reviewer again for the insightful review. If the reviewer still has questions, please feel free to discuss with us!

---

> > ### Author Response · Authors · 2024-11-25
> >
> > Dear Reviewer UsVN,
> >
> > We again thank you for your positive comments and valuable suggestions, which have been really helpful in improving our work. As the rebuttal deadline approaches, we wish to confirm whether our responses have adequately addressed your concerns.
> >
> > Regarding the theoretical aspects, we have added **Section A.2** in the appendix to specifically analyze your concerns about information loss, with results demonstrating that our method maintains acceptable levels of information preservation. We have also explained the relationship between the 2D Markov model and real-world image data in our response. Furthermore, we have supplemented our work with performance analysis on **detailed understanding tasks** and included the **non-VQA open benchmarks you suggested**. Again, we sincerely appreciate your insightful feedback, which has significantly enhanced our paper.
> >
> > **We would appreciate knowing whether these responses have fully addressed your questions and concerns. If not, we are eager to receive any further feedback you may have, especially given the approaching deadline.**
> >
> > Best regards, Authors

---

> > ### Comment · Reviewer_UsVN · 2024-11-25
> >
> > Thank you very much for your reply. I have already learned about the MME experiment. Regarding the image captioning task, has the author conducted experiments on data sets such as Nocaps and Flickr30k?

---

> ### Author Response · Authors · 2024-11-25
>
> We thank the reviewer for suggesting evaluation on specific image caption benchmarks. Given the time constraints of the rebuttal period, we just conducted tests on the Nocaps (val split) to compare LLM+VQ+BPE and LLM+VQ, analyzing how the addition of BPE affects image captioning performance. We also evaluated our version with Additional Scaling (SFT) to observe the impact of increased SFT data scaling on performance. The results are shown in the table below, reporting CIDEr, SPICE, METEOR, and ROUGE-L metrics.
>
> |                                  | CIDEr | SPICE | METEOR | ROUGE-L |
> | -------------------------------- | ----- | ----- | ------ | ------- |
> | LLM+VQ                           | 93.3  | 13.6  | 27.9   | 55.0    |
> | LLM+VQ+BPE                       | 91.5  | 13.8  | 27.4   | 53.7    |
> | LLM+VQ+BPE w/ additional scaling | 98.9  | 14.5  | 28.4   | 56.5    |
>
> We found that using BPE indeed leads to some decrease in CIDEr and ROUGE-L scores on this image captioning task. Nevertheless, as we explained earlier, considering the comprehensive improvements BPE brings across various tasks, we believe this is a meaningful trade-off.
>
> Furthermore, we observed that the version with additional scaling achieved further improvements on Nocaps, likely because the additional SFT data included some image caption instructions. Given that we did not specifically optimize instruction tuning for image captioning, we believe our method has the potential for further improvement on this task.
>
> We again thank the reviewer for suggesting evaluation on the image captioning task. Given the time constraints of the rebuttal period, we promise to include more evaluations (including Flickr30k, as suggested by the reviewer) in future revisions to more thoroughly validate and analyze our method's performance on tasks requiring detailed understanding.
>
> If the reviewer has any unresolved concerns, please feel free to discuss with us!

---

> ### Author Response · Authors · 2024-11-26
>
> Dear reviewer, we have also completed evaluation on Flickr30k (due to time constraints, we only selected a 1000-image split). Below are the evaluation results:
>
> |                                  | CIDEr | SPICE | METEOR | ROUGE-L |
> | -------------------------------- | ----- | ----- | ------ | ------- |
> | LLM+VQ                           | 76.4  | 16.2  | 24.4   | 51.3    |
> | LLM+VQ+BPE                       | 75.7  | 15.9  | 24.1   | 50.8    |
> | LLM+VQ+BPE w/ additional scaling | 80.7  | 17.2  | 25.0   | 52.6    |
>
> These results are consistent with our findings on Nocaps, showing that the addition of BPE does not lead to significant degradation in detailed understanding capabilities. As explained earlier, we reiterate that this is an expected trade-off.
>
> We hope these supplementary experiments address the reviewer's concerns, and we would greatly appreciate any further feedback from the reviewer.

---

> > ### Author Response · Authors · 2024-11-27
> >
> > Dear reviewer,
> >
> > We have conducted the experiments you suggested on Nocaps and Flickr30k, with results shown in the two tables above. Additionally, we have uploaded a new revision to include **Table G.4** in **Appendix G.2**, which provides a more intuitive comparison of results on these two benchmarks. Please feel free to check the updated PDF.
> >
> > We hope these supplementary experiments have helped you better understand our method. **If you have any unresolved concerns, please feel free to let us know! Or if our responses have addressed your questions, we would be grateful if you could consider adjusting your score accordingly.**

---

> ### Author Response · Authors · 2024-11-28
>
> Dear Reviewer,
>
> We again thank you for the time and effort you have dedicated to reviewing our paper. We particularly appreciate your recognition of our novel adaptation of BPE for images and its potential in improving visual-text alignment for multimodal models. Regarding your questions and concerns, we have addressed each point in our rebuttal above. **For your convenience, we summarize our responses as follows**:
>
> 1. **On Multimodal Fusion:**
>     - Explained that Proposition 2 shows our BPE tokenizer enables transformers to achieve near-optimal loss, naturally aligns with text tokenization, thus facilitating better modality fusion.
> 2. **On 2D Markov Process Applicability:**
>     - Explained how real images can be modeled as combinations of multiple Markov processes and formalized pixel dependencies through weighted conditional probability, capturing spatial locality and multi-scale structures.
> 3. **On Information Loss:**
>     - Derived theoretical bound on BPE information loss: $L_{bpe} \leq (|D_{bpe}| - |D_{vq}|) \times (-p_{\min}\log(p_{\min}))$
>     - Quantified maximum information loss at approximately 0.35% per image under typical settings
>     - Full analysis are included in Section A.2 of the revision.
> 4. **On Task Evaluation for Detail Understanding:**
>     - Highlighted existing evaluations on detail-sensitive tasks (OCR, Object localization) in MME and MMBench
>     - Provided additional results on:
>         - MLLM-bench for open-ended tasks
>         - Nocaps and Flickr30k for image captioning
>     - Results showed minimal performance trade-offs in detail-sensitive tasks, while maintaining advantages in most scenarios
>     - Full results are included in Section G.2 of the revision
>
> We hope our rebuttal has addressed the reviewer's concerns. **Since no new issues have been raised in recent days, if the reviewer is satisfied with our responses, we sincerely hope you would consider raising the score.**

---

> ### Author Response · Authors · 2024-11-29
>
> Dear reviewer,
>
> Following your last comment, we have conducted the additional experiments you required on the Nocaps and Flickr30k benchmark. It has been 4 days since we submitted these new results, and we are eagerly awaiting your feedback. We understand that you may be busy, but could you please just **spare a few minutes to share your current thoughts with us? We would greatly appreciate knowing whether our rebuttal has adequately addressed your questions.**

---

> ### Author Response · Authors · 2024-12-02
>
> Dear reviewer UsVN,
>
> Following your comment seven days ago requesting additional evaluations on Nocaps and Flickr30k, we promptly conducted these supplementary experiments as requested. We noticed that besides the request for additional experiments, you did not seem to have new questions regarding other points in your initial review. Additionally, since we submitted the supplementary results, there have been no subsequent comments. **Can we assume that our rebuttal has adequately addressed all your questions and concerns?** If so, we would greatly appreciate if you could adjust the score to reflect that we have addressed all your concerns.
>
> We believe that our new paradigm for training MLLMs, along with its theoretical analysis and experimental validation, **is promising and represents work worthy of being shared at ICLR**, potentially broadening research topics for more researchers. We sincerely hope you would consider a better recommendation for our work.
>
> Best regards,
>
> Authors

---

> > ### Comment · Reviewer_UsVN · 2024-12-02
> >
> > Thank you for your response. I have been busy with some things recently and haven't had a chance to respond to your message. Personally, I think a new visual encoder should focus more on extracting fine-grained visual information and narrowing the gap between the text and visual modalities. The paper is theoretically very innovative, but there are indeed some shortcomings in some tasks. I have changed my score to 8. Good luck to you.

---

> > > ### Author Response · Authors · 2024-12-02
> > >
> > > We thank the reviewer for raising the score! We appreciate the constructive feedback provided by the reviewer, and we indeed acknowledge that our BPE image tokenizer may have potential trade-offs in tasks requiring detailed understanding. In our future work, we will follow the reviewer's suggestions and attempt to minimize the losses from these trade-offs while maintaining overall performance, thereby making our method better. We again thank you for your insightful review, which has significant importance for improving our work quality and guiding future research directions.

---

### Official Review · Reviewer_e25B · 2024-11-02

**Soundness:** 2
**Presentation:** 2
**Contribution:** 2
**Rating:** 5
**Confidence:** 3

**Summary:**

This paper proposes to apply Byte-Pair Encoding to visual data, which first encode image into discrete token IDs, and then train the BPE image tokenize to get image tokens with semantic prior (e.g., previously image tokens for 'a white cat' is separated in the sequence of image tokens, after BPE, one token is representing 'a white cat'). The experiments are mainly based on applying BPE image tokenizer training to LLaMA-3.1 and compare on VQAv2, MMBench, etc.

**Strengths:**

1. This BPE image tokenization approach is novel and that potentially help the transformer better understand alignment between text and image with a semantic image token.
2. There is a theoretical analysis on how BPE tokenize benefits transformers learning in Section 3.
3. The scaling of BPE is reflected in that the model improves when adding larger scale of data such as ShareGPT4, etc.

**Weaknesses:**

1. The experimental evidences are kind of weak. First, it's far behind current MLLMs SOTA on public benchmarks. For example, the best presented number of proposed model is LLM+VQ+BPE with Additional scaling (SFT) , which achieves 60.6 on VQAv2, 44.0 on MMBench, and 48.2 on VizWiz, which is far behind similar size 7B LLaMA-based MLLMs.
2. Second, the ablation is not sufficient to show the benefit of BPE image tokenizer. Only one Table results compare LLM+VQ and LLM+VQ+BPE. The details of these two models are not illustrated, e.g., what is exactly implemented for LLM+VQ.

**Questions:**

1. The LLM+VQ+BPE is also supervised finetuned on LLaVA-One-Vision, etc data, however, is far behind LLaVA-OneVision and other models that trained on these data. Then what's the benefit of this VQ+BPE compared with previous MLLM practices?
2. Is this VQ+BPE applied to other LLM beyond LLaMA-3.1-8B and get similar observations?

---

> ### Author Response · Authors · 2024-11-22
>
> We thank the reviewer for thoughtful feedbacks and valuable suggestions for our work. To address the reviewer's concerns, we provide detailed responses below.
>
> > W1: The experimental evidences are kind of weak. First, it's far behind current MLLMs SOTA on public benchmarks. For example, the best presented number of proposed model is LLM+VQ+BPE with Additional scaling (SFT) , which achieves 60.6 on VQAv2, 44.0 on MMBench, and 48.2 on VizWiz, which is far behind similar size 7B LLaMA-based MLLMs.
>
> The performance gap between our approach and current SOTA MLLMs needs to be contextualized by considering **the vast difference in training data scales**. Current SOTA methods typically employ CLIP-based visual encoders that **benefit from extensive pretraining**. This creates an inherent advantage that isn't apparent in performance comparisons.
>
> To quantify this difference: the original CLIP model used 400 million image-text pairs [1], while later models like CLIP-ViT-L/14 (OpenCLIP series) were trained on the LAION-2B dataset, processing 32B samples over multiple epochs [2]. In contrast, our BPE Image Tokenizer was trained using just 2.78 million images, without requiring paired text captions. The fact that we achieved comparable performance to some CLIP-based MLLMs (e.g., LLaVA-1.0, Llama-Adapater-v2, etc) **using only ~0.1% of their training data** (of CLIP-based encoders) demonstrates the efficiency and potential of our approach.
>
> We want to emphasize that establishing a SOTA MLLM was not this paper's primary objective, as that would require massive data collection, computational resources, and engineering optimizations beyond this paper's scope (as acknowledged in the Limitations, Section 7). Instead, we propose a novel training paradigm for MLLMs, supported by theoretical insight and preliminary experimental validation. Our goal is to introduce a new direction for MLLM development that the research community can build upon and scale further.
>
> [1] Learning Transferable Visual Models From Natural Language Supervision. Radford et al. ICML 2021.
>
> [2] https://huggingface.co/laion/CLIP-ViT-L-14-laion2B-s32B-b82K
>
> > W2: the ablation is not sufficient to show the benefit of BPE image tokenizer. Only one Table results compare LLM+VQ and LLM+VQ+BPE. The details of these two models are not illustrated, e.g., what is exactly implemented for LLM+VQ.
>
> LLM+VQ+BPE represents our complete pipeline as proposed in Section 4. In this approach, we first use a pretrained VQ-GAN model to quantize images, then apply our trained BPE tokenizer to merge these quantized tokens, and finally feed both the processed image token IDs and text token IDs to the LLM (Llama-3.1-8B in our implementation) for learning and understanding.
>
> In contrast, LLM+VQ follows a simpler path where the image token IDs obtained from VQ-GAN quantization are directly combined with text token IDs and fed to the LLM, skipping the BPE Image tokenizer processing step. We appreciate the reviewer's suggestion for clarification, and we have added a more comprehensive description in **Section B.5** of our revision.
>
> > Q1: The LLM+VQ+BPE is also supervised finetuned on LLaVA-One-Vision, etc data, however, is far behind LLaVA-OneVision and other models that trained on these data. Then what's the benefit of this VQ+BPE compared with previous MLLM practices?
>
> As we explained earlier, our proposed VQ+BPE approach represents a fundamentally different learning paradigm from the conventional CLIP+projector method. While CLIP-based encoders require billions of pretraining samples, our MLLM with BPE Image Tokenizer achieves comparable performance with limited data. We want to emphasize that although we used (part of) the LLaVA-OneVision data, it was only for the SFT phase. This means our comparison with MLLM practices like LLaVA-OneVision maintains fairness only in the SFT stage.
>
> Regarding visual token processing, existing approaches benefit from pretrained CLIP-based encoders. Even though they didn't train these encoders from scratch, they inherently leverage the vast pretraining data advantage. Given our current computational and data resources, it's challenging to match this scale. However, we believe our results demonstrate the promise of this novel MLLM training paradigm. The performance achieved with significantly less data validates our approach and should encourage further exploration of this new framework by the research community.

---

> ### Author Response · Authors · 2024-11-22
>
> > Q2: Is this VQ+BPE applied to other LLM beyond LLaMA-3.1-8B and get similar observations?
>
> Yes, our VQ+BPE pipeline is compatible with any text-only LLM. As described in Section 4.2 (Line 324-329, Token Embedding Expansion), the only modification required is expanding the base model to accommodate the new image token IDs, followed by standard SFT procedures.
>
> In our earlier implementation, we have also tested Llama 2. However, after the release of Llama 3.1, we switched to this newer version due to its enhanced comprehension capabilities. Our method demonstrated similar observations on Llama-2-7B. To better address the reviewer's concern, we completed additional training and evaluation with the Llama 2 version. Given the time constraints of the rebuttal period, we focused only on the PT(freeze)+SFT version for comparison with the optimal Llama 3.1 configuration. The results are as follows:
>
> |                 | Training type  | VQAv2 | MMBench | MME^p  | MME^c | POPE | VizWiz |
> | --------------- | -------------- | ----- | ------- | ------ | ----- | ---- | ------ |
> | Llama2+VQ       | PT(freeze)+SFT | 54.0  | 35.7    | 991.9  | 254.2 | 75.1 | 44.4   |
> | Llama3.1+VQ     | PT(freeze)+SFT | 55.4  | 37.6    | 1054.5 | 277.0 | 76.0 | 45.3   |
> | Llama2+VQ+BPE   | PT(freeze)+SFT | 56.5  | 38.1    | 1112.2 | 277.8 | 77.9 | 44.9   |
> | Llama3.1+VQ+BPE | PT(freeze)+SFT | 57.1  | 40.9    | 1223.5 | 307.1 | 79.0 | 46.0   |
>
> The complete results table have been added to Appendix G.1 of our revision.
>
> The results demonstrate that for the Llama 2 version, using the same data and processing pipelines, the incorporation of the BPE image tokenizer similarly improved performance. While current time constraints only allowed for validation with Llama 2, we promise to expanding our evaluation to more base LLMs in the future to further demonstrate the broad applicability of our proposed method.
>
> ---
>
> If there are still some remaining concerns, we are happy to have further discussions with the reviewer.

---

> > ### Author Response · Authors · 2024-11-25
> >
> > Dear Reviewer e25B,
> >
> > We again thank you for your valuable feedback, which has greatly helped improve the quality of our paper. As the rebuttal deadline approaches, we wish to confirm whether our responses have adequately addressed your concerns. We have explained that the performance differences primarily stem from the pre-training data scale of visual encoders. We have also provided additional details in the revision regarding the pipelines for both LLM+VQ+BPE and LLM+VQ approaches. Furthermore, we have supplemented with experiments using other base LLMs to validate the applicability of our method.
> >
> > **We would appreciate knowing whether our responses have fully addressed your concerns. If not, we welcome any additional feedback you may have.**
> >
> > Best regards, Authors

---

> > > ### Author Response · Authors · 2024-11-26
> > >
> > > Dear reviewer,
> > >
> > > Thank you again for your valuable review. We have responded to your every question and concern. We hope to hear back from you! **If you have any unresolved concerns, please feel free to let us know! Or if our responses have addressed your questions, we would be grateful if you could consider adjusting your score accordingly.**

---

> > > > ### Comment · Area_Chair_BgWD · 2024-11-28
> > > >
> > > > Dear e25B,
> > > >
> > > > Could you please take a careful look at the other reviews and author responses, and comment on whether your original rating stands? Thank you.
> > > >
> > > > Best, AC

---

> ### Author Response · Authors · 2024-11-29
>
> We thank the AC for encouraging reviewers to join the discussion.
>
> We again express our gratitude for reviewer e25B's service. While we understand that the reviewer may be busy, we hope you can spare a moment to read our revisions and supplementary experiments. We sincerely wish to hear your thoughts on these responses and whether they have changed your perspective on our work.
>
> **For your convenience, here is a takeaway summary, which we hope will help you quickly grasp our key points:**
>
> We first thank the reviewer for recognizing our novel BPE tokenization approach, theoretical analysis, and demonstrated scaling benefits with larger datasets. Regarding your questions and concerns, we have responded to each as follows:
>
> 1. **On Performance Gap with SOTA:**
>     - Achieved comparable performance to some CLIP-based MLLMs while using only ~0.1% of the pre-training data for CLIP encoder (2.78M vs. over 2B images)
>     - Emphasized our contribution as a proof-of-concept for a novel training paradigm rather than pursuing SOTA performance
> 2. **On Ablation Studies:**
>     - Clarified implementation details:
>         - LLM+VQ+BPE: Complete pipeline with VQ-GAN quantization followed by BPE tokenizer
>         - LLM+VQ: Direct use of VQ-GAN tokens without BPE processing
>     - Added comprehensive descriptions in Section B.5
> 3. **On Comparison with Existing MLLMs:**
>     - Highlighted fundamental difference from CLIP+projector methods
>     - Explained that LLaVA-OneVision data was only used for SFT phase, maintaining fair comparison at that stage
> 4. **On Model Generalization:**
>     - Demonstrated compatibility with different LLMs through additional experiments on Llama 2
>     - Provided new results showing consistent performance improvements with BPE across both Llama 2 and Llama 3.1 models
>
> We hope our rebuttal has addressed the reviewer's concerns. **Given that no new issues have been raised in recent days, if the reviewer feels satisfied with our responses, we sincerely hope you would consider raising the score.**

---

> > ### Author Response · Authors · 2024-12-01
> >
> > Dear reviewer,
> >
> > With only about 2 days remaining until the rebuttal deadline, we are still eagerly awaiting your response. We sincerely hope that when you have a moment, you could spare a few minutes to check the summary above. Have your previous questions and concerns been addressed? We are very keen to know whether our rebuttal has changed your recommendation regarding our work.
> >
> > Sincerely, Authors

---

> ### Author Response · Authors · 2024-12-02
>
> Dear reviewer e25B,
>
> As we are now on the last day of the rebuttal period, and we have not received any comments about our responses, **can we assume that our rebuttal has adequately addressed all of your concerns?** If so, we would greatly appreciate if you could adjust the score to reflect that we have addressed all your concerns.
>
> We believe that our new paradigm for training MLLMs, along with its theoretical analysis and experimental validation, **is promising and represents work worthy of being shared at ICLR**, potentially broadening research topics for more researchers. We sincerely hope you would consider a better recommendation for our work.
>
> Best regards, Authors

---

> ### Author Response · Authors · 2024-12-03
>
> Dear reviewer e25B,
>
> We again thank you for your service, which helps a lot in improving our work! We believe that by addressing your questions and concerns, the quality of our work has been further improved. Do you agree that all of your concerns have been resolved? Would you be willing to adjust the score to reflect this?
>
> Considering that only about 1 hour remains for the rebuttal period, we are still waiting for your feedback and would like to know **if your original concerns still stand**. Have our efforts changed your consideration of our work? **Please at least inform us of your final conclusion.**
>
> Best regards, Authors

---

### Official Review · Reviewer_ZNHN · 2024-11-04

**Soundness:** 2
**Presentation:** 2
**Contribution:** 2
**Rating:** 5
**Confidence:** 3

**Summary:**

This paper tried to use BPE for image tokenization. From the results shown to us, there is some improvement.

**Strengths:**

1. From the results shown to us, there is some improvement.
2. Experiment settings are clear

**Weaknesses:**

1. Performance is poor compared to any CLIP style or even DINO style MLLM as the visual encoder.
2. There is no projector in the experiments. This could be an extreme unfair setting compared to classical pipeline.
3. I do not think proofs are helpful to understand what is going on in the experiments.

**Questions:**

1. Why is there a 0.99 in L770 and L253? Is this made up?
2. For Figure 2 (b) (c), all settings converge after ~150 iterations. I don't think they make any difference.
3. Why vocabulary changes from 8k-> 16k, there is a performance drop? I can not find any evidence in the proof that can demonstrate this point.
4. The proposed BPE is very similar to super pixel or clustering algorithm. Authors should discuss the difference and compare the performance.
5. In table 5.1, authors can add another classical setting: During SFT, visual encoder is frozen.

---

> ### Author Response · Authors · 2024-11-22
>
> We greatly appreciate the reviewer's valuable and constructive review on our work, which significantly improves the quality of our paper. We provide responses to each of your concerns as below.
>
> > W1: Performance is poor compared to any CLIP style or even DINO style MLLM as the visual encoder.
>
> We would like to clarify that the BPE Image Tokenizer represents a fundamentally different paradigm from conventional CLIP/DINO-style visual encoders. For meaningful performance comparisons, it is crucial to consider the vast **difference in training data scale**.
>
> The widely-used visual encoders were trained on massive datasets - the original CLIP used 400 million image-text pairs [1], while later models like CLIP-ViT-L/14 leveraged even larger datasets such as LAION-2B, processing 32B samples over multiple epochs [2]. In contrast, our BPE Image Tokenizer was trained on just 2.78 million images, without requiring paired text captions. The fact that we achieved comparable performance to some CLIP-based MLLMs (e.g., LLaVA-1.0, Llama-Adapater-v2, etc) **using only ~0.1% of their training data** (of CLIP-based encoders) demonstrates the efficiency and potential of our approach.
>
> However, we want to emphasize that establishing a SOTA MLLM was not this paper's primary objective, as that would require massive data collection, computational resources, and engineering optimizations beyond this paper's scope (as acknowledged in the Limitations, Section 7). Instead, our contribution lies in proposing a novel training paradigm for MLLMs, supported by theoretical insight and preliminary experimental validation. We believe this opens up a promising new direction for the MLLM research community to explore.
>
> [1] Learning Transferable Visual Models From Natural Language Supervision. Radford et al. ICML 2021.
>
> [2] https://huggingface.co/laion/CLIP-ViT-L-14-laion2B-s32B-b82K
>
> > W2: There is no projector in the experiments. This could be an extreme unfair setting compared to classical pipeline.
>
> We would like to point out that our BPE Image Tokenizer directly processes images into token IDs (as described in Section 4.1, Line 306-308). These token IDs are integers, functionally equivalent to text token IDs in LLMs, and they are jointly fine-tuned during the SFT phase. Similar to how text LLMs map tokens to embedding layer indices in transformer models, our approach **directly maps image token IDs to corresponding embedding indices**. Therefore, a projector is neither necessary nor applicable in our framework, as we achieve modality alignment through direct token-level integration rather than feature-space projection.
>
> > W3: I do not think proofs are helpful to understand what is going on in the experiments.
>
> We believe Section 3 provides clear and intuitive theoretical support for our algorithm design. Our theoretical findings can be summarized as follows: Using a token merging mechanism similar to text-based BPE algorithms can achieve near-optimal performance even in worst-case scenarios, while approaches without such merging may suffer from significant entropy loss. This indicates that discretization followed by BPE token merging can significantly enhance a transformer model's understanding of two-dimensional sequences like images.
>
> This theoretical insight directly guided our method design, and we validated it through:
>
> 1. A toy experiment (Figure 2) that empirically demonstrates the theoretical findings.
> 2. A complete MLLM training pipeline that implements these insights at certain scale.
>
> We maintain consistency between theory and practice throughout the paper - providing clear theoretical insights while experimentally validating their practical implications. This coherent progression from theoretical insight to practical implementation helps readers understand both why and how our approach works.
>
> > Q1: Why is there a 0.99 in L770 and L253? Is this made up?
>
> We would like to point out that the 0.99 in line 253 is based on Lemma A.1 (line 770), which we **cite from prior work** [3] as indicated in line 759. This is not an arbitrary number.
>
> To provide further context: in the original proof of Lemma A.1, the value 0.99 is used to represent a constant arbitrarily close to 1 (as explained in the footnote on page 9 of [3]). We maintain this notation for consistency with the cited work. Therefore, this value has mathematical significance within the theoretical framework rather than being an arbitrary choice.
>
> [3] Toward a Theory of Tokenization in LLMs. Rajaraman et al. 2024.

---

> ### Author Response · Authors · 2024-11-22
>
> > Q2: For Figure 2 (b) (c), all settings converge after ~150 iterations. I don't think they make any difference.
>
> We appreciate this observation and would like to clarify an important visualization issue in Figure 2(b)(c):
>
> The y-axis of these figures were different in the original paper version, as we attempted to scale Figure 2(c) to maximize plot visibility. This scaling may have obscured the key difference between the two scenarios. We have now **adjusted Figure 2(c) to use the same scale as Figure 2(b) in the revision**, which reveals a crucial distinction:
>
> In Figure 2(b), the transformer model without tokenizer consistently converges to a suboptimal value (dotted line), maintaining a significant gap from the optimal cross-entropy loss (dashed line). In contrast, Figure 2(c) shows that models using a tokenizer can easily achieve the optimal cross-entropy loss. While both approaches show similar convergence rates, their final performance differs substantially - the model with tokenizer achieves meaningfully better loss values.
>
> Thank you for highlighting this visualization issue. The rescaled figures should now better illustrate that despite similar convergence speeds, the final performance differs significantly between the two approaches.
>
> > Q3: Why vocabulary changes from 8k-> 16k, there is a performance drop? I can not find any evidence in the proof that can demonstrate this point.
>
> According to Proposition 2, increasing the vocabulary size (D) reduces $\varepsilon = \log(1/\delta)/(0.99\log(D))$, which then reduces the bound $\frac{1}{1-\varepsilon}H_{\infty}$. However, this theoretical improvement exhibits diminishing returns: when $\varepsilon$ is already small, further reductions in $\frac{1}{1-\varepsilon}$ become marginal.
>
> Meanwhile, increasing vocabulary size introduces practical challenges: the transformer model requires larger embedding sizes to accommodate more tokens, which can complicate the training process and potentially impact model performance. This creates a trade-off - while larger vocabularies might offer marginal theoretical improvements, they also increase model complexity and training difficulty.
>
> Given the current limitations in theoretical understanding of transformer models, it's challenging to provide a complete theoretical explanation for this trade-off. As noted in lines 470-474 of our paper, we can only offer intuitive explanations for the observed performance drop when vocabulary size increases from 8k to 16k.
>
> > Q4: The proposed BPE is very similar to super pixel or clustering algorithm. Authors should discuss the difference and compare the performance.
>
> We appreciate the reviewer's insightful observation about the relationship between our BPE Image Tokenizer and superpixel/clustering algorithms. Indeed, there are meaningful similarities between these approaches, as they all aim to group visual elements into meaningful units. We would like to clarify the key distinctions and contributions of our approach:
>
> 1. Learning Objective: Our tokenizer learns to merge tokens based on statistical co-occurrence patterns in the training data, optimizing specifically for language model understanding. This differs fundamentally from clustering/superpixel methods that optimize for visual space similarity metrics.
> 2. Multimodal Integration: Our approach is specifically designed to align with language model training paradigms. By adopting a BPE-inspired method, we create a natural bridge between visual and textual modalities in transformer architectures. Unlike clustering methods that operate in continuous feature spaces, our tokenizer works directly with discrete token indices from VQ-GAN, enabling seamless integration without additional projection layers.
>
> Regarding performance comparisons, a direct ablation study would be challenging due to fundamental pipeline differences. Traditional clustering approaches require continuous feature space computations and additional projection layers for transformer compatibility. Moreover, as these methods typically rely on CLIP-based encoders trained on substantially larger datasets, fair performance comparisons would be difficult to establish.
>
> While we acknowledge the value of comparative studies, our paper's primary contribution is introducing a novel MLLM training paradigm supported by theoretical analysis and preliminary validation. A comprehensive evaluation using larger-scale training and broader comparisons remains an important direction for future work.

---

> ### Author Response · Authors · 2024-11-22
>
> > Q5: In table 5.1, authors can add another classical setting: During SFT, visual encoder is frozen.
>
> Thank you for this suggestion. We have conducted additional experiments on the LLM+VQ+BPE setting by freezing the embeddings corresponding to visual tokens during the SFT phase. The results are shown as below:
>
> | Training type                      | VQAv2 | MMBench | MME^p  | MME^c | POPE | VizWiz |
> | ---------------------------------- | ----- | ------- | ------ | ----- | ---- | ------ |
> | SFT                                | 52.2  | 35.4    | 1029.7 | 269.6 | 76.3 | 45.3   |
> | PT(full)+SFT                       | 56.5  | 38.6    | 1144.6 | 284.3 | 77.3 | 45.8   |
> | PT(freeze text)+SFT                | 57.1  | 40.9    | 1223.5 | 307.1 | 79.0 | 46.0   |
> | PT(full)+SFT(freeze visual)        | 31.5  | 17.8    | 624.1  | 171.9 | 46.4 | 29.5   |
> | PT(freeze text)+SFT(freeze visual) | 22.5  | 13.3    | 488.7  | 143.6 | 35.2 | 21.5   |
>
> In the results, we use "(freeze text)" and "(freeze visual)" to distinguish which token embeddings were frozen during the PT and SFT phases. The results reveal that freezing embeddings during SFT leads to significant performance degradation. This is expected since our approach unifies both image and text modalities into index tokens - the transformer model needs to learn to understand both token types simultaneously during SFT to properly process multimodal inputs during inference.
>
> Interestingly, when visual tokens are frozen, PT(full) slightly outperforms PT(freeze text). We hypothesize that this occurs because the PT phase with both visual and text token fine-tuning provides a limited form of modality alignment, partially compensating for the lack of full SFT. This offers marginally better results compared to versions without any text-visual alignment training.
>
> These findings further support our framework's design principle of unified token-level learning across modalities. We have also included a full table with these results in Appendix G.1 in the revision PDF.
>
> ---
>
> We hope the response resolves the reviewer's concerns. If the reviewer still feels there're something unclear, we're happy to have further discussions!

---

> > ### Author Response · Authors · 2024-11-24
> >
> > Dear Reviewer ZNHN,
> >
> > We again thank you for your valuable feedback on our paper. As the discussion deadline approaches, we wish to confirm whether our responses have adequately addressed your concerns. We have clarified that the main gap between our approach and existing methods lies in the amount of pre-training data used - ie, we did not utilize large amounts of pre-training data like CLIP-based encoders. Furthermore, we have emphasized that this paper's scope focuses on exploring a novel training paradigm rather than engineering a state-of-the-art MLLM. Additionally, we have also provided supplementary experimental results and revisions in response to your concerns.
> >
> > **We would appreciate knowing whether our responses have fully addressed your concerns. If not, we are eager to receive further feedback from you!**
> >
> > Best regards,
> >
> > Authors

---

> > > ### Author Response · Authors · 2024-11-26
> > >
> > > Dear reviewer,
> > >
> > > We again thank you for your time and efforts. We have responded to each of your questions and concerns, and we look forward to hearing back from you! **If you have any unresolved concerns, please feel free to let us know. Or if our responses have adequately addressed your questions, we would be grateful if you could consider adjusting your score accordingly.**

---

> > > > ### Comment · Area_Chair_BgWD · 2024-11-28
> > > >
> > > > Dear ZNHN,
> > > >
> > > > Could you please take a careful look at the other reviews and author responses, and comment on whether your original rating stands? Thank you.
> > > >
> > > > Best, AC

---

> ### Author Response · Authors · 2024-11-29
>
> We thank the AC for encouraging the reviewer to join the discussion.
>
> We again express our gratitude for reviewer ZNHN's service, and while we understand that the reviewer may be busy, we sincerely hope you can spare a moment to read our responses and revisions. We sincerely wish to hear your thoughts on these responses and whether they have changed your perspective on our work.
>
> We first appreciate your recognition of our experimental improvements and clarity in experimental settings. Regarding your questions and concerns, we have addressed each point in our rebuttal above. **For your convenience, here is a takeaway summary, which we hope will help you quickly grasp our key points**:
>
> 1. **On Comparison with CLIP-based MLLMs:**
>     - Achieved comparable performance to some CLIP-based MLLMs while using only ~0.1% of the pre-training data for CLIP encoder (2.78M vs. over 2B images)
>     - Emphasized our contribution as a proof-of-concept for a novel training paradigm rather than pursuing SOTA performance
> 2. **On Projector Absence:**
>     - Clarified that our approach directly maps image token IDs to embedding indices, similar to text tokens in LLMs
>     - Explained why projector is unnecessary in our framework as modality alignment happens at token level
> 3. **On Theoretical Framework:**
>     - Connected theory to practice through:
>         - Theoretical proof showing BPE-style merging achieves near-optimal performance
>         - Empirical validation via toy experiments and full MLLM pipeline
> 4. **On Technical Questions:**
>     - Clarified that 0.99 constant comes from cited prior work
>     - Addressed visualization issues in Figure 2
>     - Explained performance drop with larger vocabulary (8k->16k) as trade-off between theoretical improvement and practical challenges in transformer
>     - Distinguished our BPE approach from superpixel/clustering methods through learning objectives and multimodal integration
> 5. **On Additional Experiments:**
>     - Provided new results with frozen visual encoder during SFT
>     - Results showed significant performance degradation with frozen embeddings, supporting our unified token-level learning design
>
> We hope our rebuttal has addressed the reviewer's concerns. **Given that no new issues have been raised in recent days, if the reviewer feels satisfied with our responses, we sincerely hope you would consider raising the score.**

---

> > ### Author Response · Authors · 2024-12-01
> >
> > Dear reviewer,
> >
> > With only 2 days remaining until the rebuttal deadline, we are still eagerly awaiting your response. We sincerely hope that when you have a moment, you could spare a few minutes to check the summary above. Have your previous questions and concerns been addressed? We are very keen to know whether our rebuttal has changed your recommendation regarding our work.
> >
> > Sincerely, Authors

---

> ### Author Response · Authors · 2024-12-02
>
> Dear reviewer ZNHN,
>
> As we are now on the last day of the rebuttal period, and we have not received any comments about our responses, **can we assume that our rebuttal has adequately addressed all of your concerns?** If so, we would greatly appreciate if you could adjust the score to reflect that we have addressed all your concerns.
>
> We believe that our new paradigm for training MLLMs, along with its theoretical analysis and experimental validation, **is promising and represents work worthy of being shared at ICLR**, potentially broadening research topics for more researchers. We sincerely hope you would consider a better recommendation for our work.
>
> Best regards, Authors

---

> > ### Comment · Reviewer_ZNHN · 2024-12-03
> > **Thank you for your efforts!**
> >
> > I truly appreciate authors' response.
> >
> > My concern with the connector is that VQ typically capture the low level vision information, while LLM is good at semantical knowledge. Thus, directly putting VQ+LLM is not optimal. However, the role of connector is to bridge this gap, which have been demonstrated a lot of times in VLM training literatures.

---

> > > ### Author Response · Authors · 2024-12-03
> > >
> > > We appreciate the reviewer's response. We would like to kindly remind the reviewer that **directly combining VQ+LLM is not our approach - this was only used as an ablation baseline**. Instead, we designed a VQ+**BPE**+LLM approach that achieves a unified representation of text and images, thereby enabling better connection between LLM and VQ. **The design of the BPE Image tokenizer is our main intended contribution**. We hope the reviewer can reconfirm this point. Given the very limited time for rebuttal, if the reviewer has any questions about our clarification, please feel free to raise them, and we will respond immediately.

---

> ### Author Response · Authors · 2024-12-03
>
> We would like to further explain that the traditional methods you mentioned follow a pipeline of **image -> CLIP features -> connector -> LLM embeddings**, which typically applies pre-trained CLIP encoders to images to obtain CLIP features, then uses a connector (usually MLP networks) to map to LLM embedding layers.
>
> In contrast, our approach follows **image -> VQ IDs -> 2D-BPE processing -> direct input to LLM together with text IDs**. Our method is totally different from the approaches you mentioned and represents a completely new paradigm. In our approach, images are directly processed into integer IDs, then BPE is used to achieve **early-fusion** of information from the image, which **directly corresponds to LLM embedding dimensions** (just like text IDs), rather than using MLP connectors for **late-fusion** as in traditional methods. The traditional approach **has some shortcomings in aligning image and text modalities**, and the reviewer can refer to the first paragraph of our introduction (lines 028-038) for more detailed description and citations of this, which is precisely the problem our work aims to solve.
>
> We again hope the reviewer can check the distinction between our method and traditional connector-based approaches. We believe this is an interesting new paradigm that achieves promising performance **using only ~0.1% of the pre-training data compared to CLIP-based encoders**. We believe this is work worth sharing and discussing at ICLR.

---

> ### Author Response · Authors · 2024-12-03
>
> Dear reviewer,
>
> We hope our above response has clarified the fundamental differences between our method and the connector-based methods you mentioned. While these methods have been demonstrated in some VLM training literature, several works have also discussed **their shortcomings in cross-modal fusion** (as we cited in the first paragraph of our introduction [1, 2, 3]). Therefore, this is not yet an optimal paradigm, and **the research community still needs to discuss potential optimizations**. In our paper, we demonstrate **both theoretically and experimentally** the performance improvements brought by VQ+BPE+LLM. We sincerely hope the reviewer will reconsider our work's contribution.
>
> ---
>
> [1] Multimodal machine learning: A survey and taxonomy. Baltrusaitis et al.
>
> [2] Chameleon: Mixed-modal early-fusion foundation models. Chemeleon Team.
>
> [3] Unified language-vision pretraining with dynamic discrete visual tokenization. Jin et al.

---

> ### Author Response · Authors · 2024-12-03
>
> Dear reviewer ZNHN,
>
> We appreciate your constructive review. Considering that **most concerns in your original review have been properly addressed, especially:**
>
> - the 0.99 in the theorems
> - the differences in Figure 2(b) & 2(c)
> - the theoretical evidence for performance changes in vocabulary from 8k to 16k
> - supplementing experiments to verify what happens when freezing the visual part during the SFT stage
>
> We hope that resolving these questions has helped the reviewer better understand why and how our method can bring improvement. Based on the above, **we believe that the reviewer's original concerns and misunderstandings about the effectiveness of our method should have been cleared up. Could you please consider updating your score to reflect this?**
>
> Best regards, Authors

---

> ### Author Response · Authors · 2024-12-03
>
> Dear reviewer ZNHN,
>
> We thank you again for your insightful review, which helps a lot in improving our work! For your latest comment, we responded to it **immediately within the following few minutes** and subsequently provided more information. We hope you have seen our new responses.
>
> - Do you have any comments on our clarification above regarding the main differences between our method and the connector-based methods you mentioned, as well as the shortcomings we aim to address?
> - Also, do you agree that your main initial concerns (as listed in the points above) have been resolved? Would you be willing to adjust the score to reflect this?
>
> Considering that only about 1 hour remains for the rebuttal period, we are still waiting for your feedback and would like to know **if your original questions still stand**. Have our efforts changed your consideration of our work? **Please at least inform us of your final conclusion.**
>
> Best regards, Authors

---

### Author Response · Authors · 2024-11-22
**Global response to all reviewers**

We thank all the reviewers for the efforts in reviewing our paper and providing insightful suggestions! This has been of great help in improving our paper. In response to the concerns raised by the reviewers, we have provided detailed replies in the responses below.

We have also conducted additional experiments and made revisions to the paper based on the reviewers' suggestions. The newly added content is highlighted in blue text within the revised version. Specifically, we've made the listed revisions:

- Adjusted the y-axis scale in Figure 2(c) to better align with Figure 2(b), preventing potential misunderstanding of the two figures.
- Following reviewer UsVN's suggestion, added Section A.2 to discuss the information loss of our BPE image tokenizer.
- Following reviewer e25B's suggestion, added Section B.5 to provide more detailed descriptions of both LLM+VQ+BPE and LLM+VQ pipelines.
- Following reviewer ZNHN's suggestion, included experiments on freezing visual tokens during the SFT stage, with results presented in Section G.1.
- Following reviewer e25B's suggestion, included experiments using different base LLM (Llama 2), with results presented in Section G.1. Additionally, supplemented extra evaluation results analyzing our method's performance in handling detailed information, with results presented in Section G.2.

Feel free to check the updated PDF paper. If there are still questions, please let us know. We are looking forward to further discussion!

---

> ### Author Response · Authors · 2024-11-28
>
> Dear Reviewers,
>
> As the deadline for uploading the revised PDF approaches, we have made the following new updates based on our responses and experiments:
>
> - Following reviewer UsVN's latest suggestions, we have completed evaluations on image captioning tasks (Nocaps & Flickr30k) and included the results in **Table G.4** of the revision
> - To better address the performance concerns raised by reviewers ZNHN and e25B, we have added **Section. H** to more formally clarify that traditional MLLMs using CLIP-based encoders have an implicit advantage in terms of pre-training data, and that the scope of our paper is to explore a proof-of-concept, which is a common approach in the research community.
>
> While we believe our revisions have addressed the reviewers' concerns and strengthened our work, we sincerely hope to **hear from the reviewers for a more objective evaluation**, as this would be invaluable for further improving the quality of our work. We would greatly appreciate it if the reviewers could **share their thoughts on the revisions and let us know whether our rebuttal has adequately addressed their concerns.**

---

> > ### Author Response · Authors · 2024-12-01
> >
> > Dear reviewers
> >
> > We understand that you must have a very busy schedule, and we truly appreciate the time you've already dedicated to reviewing our paper. Your insights have been invaluable to improving our work. We noticed that we haven't received your response to our recent responses, and we're eager to move forward with your feedback. Given the approaching deadline, would it be possible for you to provide your feedback at your earliest convenience? **We would be grateful for even brief comments.** Thank you again for your expertise and consideration.
> >
> > Sincerely, Authors

---

> > > ### Author Response · Authors · 2024-12-02
> > >
> > > Dear reviewers,
> > >
> > > We hope this message finds you well. We would like to remind you that the author-reviewer discussion phase **will end in about one day**, and we have been always awaiting your responses. Have our rebuttals adequately addressed your questions and concerns? Has your consideration of our paper changed? We sincerely look forward to hearing your thoughts!
> > >
> > > Sincerely, Authors

---

### Comment · Area_Chair_BgWD · 2024-11-24

Dear Reviewers,

This is a friendly reminder that the discussion period will end on Nov 26th (Anywhere on Earth). If you have not already, please take a careful look at the other reviews and author responses, and comment on whether your original rating stands. Thank you.

Best, AC

---

> ### Author Response · Authors · 2024-12-03
>
> Dear reviewer ZNHN & e25B,
>
> This is a friendly reminder that only **half a day** remains in the rebuttal period, after which authors and reviewers will no longer be able to communicate. We are still eagerly awaiting your response. We would like to **confirm whether you agree that our responses and revisions have adequately addressed all of your questions and concerns.** Would you be willing to take a moment to check our rebuttal and share your thoughts?
>
> Best regards, Authors

---

### Comment · Area_Chair_BgWD · 2024-11-28

Dear reviewers,

This is a friendly reminder that the discussion period has been extended until December 2nd. If you haven’t yet, we kindly encourage you to review the authors' rebuttal and messages at your earliest convenience and confirm whether your comments have been adequately addressed.

We greatly appreciate your service to this process.

Best, AC

---

### Comment · Area_Chair_BgWD · 2024-12-03

Dear reviewers ZNHN & e25B,

Could you please review the authors' rebuttal and messages and confirm whether your comments have been adequately addressed?  There are only a few hours left in the discussion period, and your input would be much appreciated.  Thank you.

Best,
AC

---

### Author Response · Authors · 2024-12-03
**Summary of the rebuttal**

Dear AC and reviewers,

We sincerely thank you for the time and effort you have dedicated to reviewing our work. We greatly appreciate all reviewers' insightful comments and constructive suggestions. In particular, we would like to express our gratitude to reviewer UsVN for the valuable suggestion regarding information loss. This inspired us to supplement our revision with additional theoretical proofs and discussions on information loss, which have further strengthened the theoretical support of our work. We are also grateful to all reviewers for their meticulous questions, which helped us identify potential confusions or misunderstandings for readers. The revisions made based on these reviews have improved the quality of our work.

The core motivation of our research lies in our attempt to achieve unified representations of cross-modal information through BPE-style processing, incorporating structural priors into tokens to enable early-fusion of modal information. We believe this design represents an innovative step forward from traditional MLLM training pipelines, and we are confident that it will spark interest among researchers and inspire follow-up work in this direction.

While we understand that some reviewers may have been too busy during the rebuttal period, which resulted in limited discussion during the rebuttal period, we still **encourage more discussion during the subsequent AC-reviewer discussion phase** to confirm whether our rebuttal has adequately addressed reviewers' questions and concerns.

For your convenience, **to help the AC and reviewers more easily grasp the key points of the entire rebuttal, we provide a summary here**, hoping everyone can have a better understanding.

---

# Our work in brief

We propose a novel BPE Image Tokenizer that applies byte-pair encoding (BPE) principles to visual data, enabling better integration of visual information into MLLMs. Unlike conventional approaches using separate visual encoders, our method directly incorporates structural prior information into image tokens. We provide theoretical analysis showing why this paradigm benefits transformers' learning of 2D sequence data, and validate it through comprehensive experiments.

---

# Reviewers' positive recognitions

The reviewers have recognized several aspects of our work:

- **From reviewer ZNHN:**
	- Clear experimental settings
	- Observable improvements in the presented results
- **From reviewer e25B:**
	- The novelty of the BPE image tokenization approach that could help transformers better understand text-image alignment
	- The theoretical analysis demonstrating BPE's benefits for transformer learning
	- The demonstrated model improvements when scaling with larger training data
- **From reviewer UsVN:**
	- An innovative approach to adapting byte-pair encoding (BPE) for images
	- The method's potential in facilitating text-image alignment through structural information integration

---

> ### Author Response · Authors · 2024-12-03
>
> # Summary of our responses
>
> ---
>
> **For Reviewer ZNHN:**
>
> 1. **Regarding performance comparison with CLIP based MLLMs**
>
> - Clarified that our method achieved comparable results to some CLIP-based MLLMs, despite using significantly less pre-training data in our BPE image tokenizer compared to CLIP encoders (e.g., 2.78M vs 2B+ images for CLIP-ViT-L/14)
> - Emphasized that the paper's primary contribution lies in exploring a novel training paradigm rather than pursuing SOTA performance, which is outside the scope of this work.
>
> 2. **About why not using projector**
>
> - Explained that we directly process images into discrete token IDs, which is fundamentally different from the traditional pipeline that uses a projector to map continuous features into embeddings. Using a projector is not feasible in our framework.
> - Therefore, we cannot directly conduct a specific ablation comparison between our method and the traditional pipeline regarding the connector/projector.
>
> 3. **Theoretical Concerns**
>
> - About the relationship between our theory and experiments:
>     - We proved the near-optimal performance with BPE-style merging
>     - Then validated via toy experiments
>     - Then we built complete MLLM training pipeline to further validate
> - Clarified that the 0.99 constant comes from cited prior work
> - Provided mathematical justification for vocabulary size impact on performance
>
> 4. **Visualization and Results Interpretation**
>
> - Addressed visualization issues in Figure 2 by adjusting scales in y-axis for better comparison
> - Explained how the trade-off happens when vocabulary size changes
>
> 5. **Additional Experimental Results**
>
> - Conducted new experiments with frozen visual encoder during SFT
> - Provided comprehensive results showing how different freezing strategies affect performance
>
> ---
>
> **For reviewer e25B:**
>
> 1. **Performance Gap with SOTA Models**
>
> - *Same as our first response to reviewer ZNHN*
>
> 2. **Ablation Study and Implementation Details**
>
> - Provided comprehensive clarification of model implementations:
>     - LLM+VQ+BPE: Complete pipeline using pretrained VQ-GAN for quantization, followed by BPE tokenizer processing
>     - LLM+VQ: Direct combination of VQ-GAN tokens with text tokens, without BPE processing
> - Added detailed descriptions in Section B.5 of the revision
>
> 3. **Comparison with Existing MLLMs (e.g., LLaVA-OneVision)**
>
> - Explained that despite using similar data for SFT, LLaVA-OneVision has huge implicit advantage of pre-training data since it uses CLIP encoder
> - Emphasized the fundamental difference from conventional CLIP+projector methods
> - Highlighted the efficiency of achieving comparable performance with significantly less pretraining data
>
> 4. **Model Generalization Beyond LLaMA-3.1**
>
> - Demonstrated broader applicability through additional experiments with Llama 2
> - Provided comprehensive comparison results, validated that the benefits of VQ+BPE generalize across different base models
>
> ---
>
> **For reviewer UsVN:**
>
>
> 1. **Addressed multimodal fusion concerns**
>
> - Demonstrated how Proposition 2 provides performance bounds for transformer learning
> - Explained how BPE tokenizer design naturally aligns with text tokenization
>
> 2. **Clarified 2D Markov process applicability**
>
> - Formalized how real images can be modeled as combinations of multiple Markov processes
> - Explained how the model captures spatial locality and multi-scale dependencies
>
> 3. **Added comprehensive information loss analysis**
>
> - Supplemented the proof of theoretical bound for information loss
> - Using the bound, demonstrated a maximum information loss of ~0.35% per image in our experimental setting
> - Added detailed analysis in Section A.2 of the revision
>
> 4. **Performance on Detail-Sensitive Tasks**
>
> - Provided detailed breakdowns of MME benchmark subcategories showing:
>     - Strong performance in most perception tasks
>     - Only minor trade-offs in detail-heavy tasks like OCR
> - Conducted additional evaluations on:
>     - MLLM-bench for open-ended tasks
>     - Nocaps and Flickr30k for image captioning
>     - Results showed acceptable performance trade-offs while maintaining advantages in most scenarios

---

### Meta-Review · Area_Chair_BgWD · 2024-12-20

**Metareview:**

This paper proposes a new image tokenizer, BPE Image Tokenizer, which merges image token IDs to enhance the incorporation of visual information into MLLMs.  The paper initially received scores of 5,5,6.  Strengths include novel approach, theoretical analysis, and some promising results.  Weaknesses include relatively weak performance compared to state-of-the-art, some limitations in the theoretical framework, and issues with ablation study.  The rebuttal addressed several of these concerns, and the final score was 5,5,8.  Despite multiple requests by the AC, there was little discussion provided by some of the reviewers, including no participation from one of the 5 reviewers.  The other 5 reviewer, during the AC-reviewer discussion phase, notified to the AC that their score is between a 5 and 6 and would not be surprised if the paper is accepted.  The AC carefully reviewed the paper, rebuttal, messages, and feel that despite some shortcomings in the empirical results, the approach is interesting and can bring a novel perspective to the MLLM literature.  Overall, the AC feels that the strengths outweigh the weaknesses, and recommends accept.  Please incorporate all of the promised revisions into the final version.

**Additional Comments On Reviewer Discussion:**

Strengths include novel approach, theoretical analysis, and some promising results.  Weaknesses include relatively weak performance compared to state-of-the-art, some limitations in the theoretical framework, and issues with ablation study.  The rebuttal addressed several of these concerns, and the final score was 5,5,8.  Despite multiple requests by the AC, there was little discussion provided by some of the reviewers, including no participation from one of the 5 reviewers.  The other 5 reviewer, during the AC-reviewer discussion phase, notified to the AC that their score is between a 5 and 6 and would not be surprised if the paper is accepted.  The AC carefully reviewed the paper, rebuttal, messages, and feel that despite some shortcomings in the empirical results, the approach is interesting and can bring a novel perspective to the MLLM literature.  Overall, the AC feels that the strengths outweigh the weaknesses, and recommends accept.

---

### Decision · Program_Chairs · 2025-01-22

Accept (Poster)